# Krylov complexity of many-body localization: Operator localization in Krylov basis

Fabian Ballar Trigueros[1] and Cheng-Ju Lin[1*]

**1** Perimeter Institute for Theoretical Physics, Waterloo, Ontario, Canada N2L 2Y5

* cjlin@pitp.ca

June 17, 2022

## Abstract

We study the operator growth problem and its complexity in the many-body localization (MBL) system from the Lanczos algorithm perspective. Using the Krylov basis, the operator growth problem can be viewed as a single-particle hopping problem on a semi-infinite chain with the hopping amplitudes given by the Lanczos coefficients. We find that, in the MBL systems, the Lanczos coefficients scale as $\sim n/\ln(n)$ asymptotically, same as in the ergodic systems, but with an additional even-odd alteration and an effective randomness. We use a simple linear extrapolation scheme as an attempt to extrapolate the Lanczos coefficients to the thermodynamic limit. With the original and extrapolated Lanczos coefficients, we study the properties of the emergent single-particle hopping problem via its spectral function, integrals of motion, Krylov complexity, wavefunction profile and return probability. Our numerical results of the above quantities suggest that the emergent single-particle hopping problem in the MBL system is localized when initialized on the first site. We also study the operator growth in the MBL phenomenological model, whose Lanczos coefficients also have an even-odd alteration, but approach constants asymptotically. The Krylov complexity grows linearly in time in this case.

# 1   Introduction

Motivated by the recent developments of quantum many-body chaos, characterizing how an operator grows and its complexity under a unitary many-body dynamics in the Heisenberg picture has been a research focus across different subfields of physics [1–6]. A popular characterization of the growth of an operator is by the out-of-time-ordered commutators (correlators), which emphasizes the operator growth in the space-time picture [7–17]. Another characterization is via the operator entanglement, which has also been studied in various systems [18–24].

Recently, Parker *et al.* [25] proposed a different characterization of operator growth from the perspective of the recursive method (or Lanczos algorithm). The Lanczos algorithm generates a set of basis for the Krylov subspace (we dub it as Krylov basis), which is used to represent the operator. The dynamics is therefore encoded in the time-dependent coefficients of the basis. In a sense which we will make precise later, the operator growth problem in this basis can be viewed as the dynamics of a single-particle wavefunction hopping on a semi-infinite chain with the hopping amplitudes given by the Lanczos coefficients generated by the Lanczos algorithm, initialized on the first site. One can therefore define various complexity measures from this point of view. For example, Ref. [25] defines the Krylov complexity to be the mean position of the emergent single particle. Ref. [26,27] propose Krylov entropy to measure the degree of spreading of the emergent single-particle wavefunction on the semi-infinite chain.

Ref. [25] hypothesizes that, for a chaotic system, the Lanczos coefficients grow as fast as possible asymptotically. For a local Hamiltonian, the Lanczos coefficients are upper bounded by a function linearly in the Krylov order asymptotically (with a logarithmic correction in 1D), which implies that the Krylov complexity grows exponentially in time (or grows as a stretched exponential in time in 1D). In contrast, in the integrable models, the Lanczos coefficients grow as a square root of the Krylov order or as a constant [25, 28]. Krylov complexity is also studied with some general considerations [29, 30] and studied in conformal field theories [31], in models with holographic correspondence [27] and in models with strong and almost strong edge modes [32–34]. The delocalization of the operators in the Krylov space has also been studied in a quantum network [35].

In this paper, we study the operator growth and the Krylov complexity in a many-body localized (MBL) system [36–43] to contrast and complement the previous results in the chaotic and integrable models. Operator growth in the MBL systems has been studied using the characterization of out-of-time-ordered correlators and it is found that the operators grow with a logarithmic "operator cone" in space-time [44–50]. Refs. [22, 23] study the operator entanglement in a MBL system, finding a logarithmic entanglement growth in time. In Ref. [51], a lower bound on the

Lanczos coefficients is derived for a many-body localized spin chain. Surprisingly, such a lower bound is also shared with a generic chaotic spin chain. Accordingly, the operator growth problem in the MBL system in the Krylov basis and its Krylov complexity certainly warrants further studies.

Particularly, we study the operator growth problem in a quantum Ising model with a longitudinal field and random transverse fields [40, 41, 52]. We find that, while the Lanczos coefficients in the MBL systems have the same asymptotic behavior as in the chaotic systems, there is an additional even-odd alternation and an apparent effective randomness. By numerically calculating the evolution of the emergent single-particle wavefunction hopping on the semi-infinite chain, we find that the Krylov complexity is bounded in time in the MBL systems. Furthermore, we also find that the emergent single-particle hopping problem is localized when initialized on the first site, and therefore the operator is localized in the Krylov subspace.

It is worth noting that several works recently have raised skepticism in the existence of MBL in the thermodynamic limit, such as the challenges from the spectral form factor [53], the growth of number entropy [54], and the obstruction from the spectral function [55]. These challenges prompt the needs for a more thorough investigating of the MBL system itself, especially by understanding the many-body resonances in MBL and their manifestations [56–58]. While the phenomena of MBL at finite sizes are relatively established, extrapolating them to the thermodynamic limit is indeed very subtle and intricate. Here in this work, we also attempt to extrapolate the behavior of the Lanczos coefficients in the MBL systems to the thermodynamic limit, using the simplest linear extrapolation [59], and discussing several results from such an extrapolation. We note that the linear extrapolation could be too simple and likely will not reflect the true thermodynamic limit. More sophisticated extrapolation procedures are likely required and our extrapolation scheme is a first step to this end.

In addition to the microscopic MBL model, we also study the operator growth problem in the MBL phenomenological model [60–64], where the initial operator is a "$l$-bit flipping" operator. We find the Lanczos coefficients in this case again show an even-odd alteration but approach constants asymptotically. We find that the Krylov complexity grows linearly in this case, and the time evolution of the emergent single-particle wavefunction has a traveling wavefront propagating linearly in time.

This paper is organized as follows. In Sec. 2, we lay the framework used to study the operator growth using Krylov basis and define different functions of interests including the Krylov complexity. We then discuss the results of operator growth in a MBL system realized in a quantum Ising model with random fields in Sec. 3, where we show the Lanczos coefficients and the linear-extrapolated results. Using the original and the extrapolated Lanczos coefficients, we then study various quantities associated with the emergent single-particle hopping problem, including the spectral functions, zero modes, Krylov complexity, wavefunction profile and return probability. In Sec. 4, we study the operator growth in the MBL phenomenological model by first demonstraing the behavior of its Lanczos coefficients. We then calculate its Krylov complexity and the wavefunction profile. We conclude in Sec. 5 with some discussion of our results, open questions and future directions.

## 2    Setup

We begin by setting up the framework to study the operator growth using the Krylov basis. Consider a quantum many-body system described by a Hamiltonian $H$ and an initial local Hermitian

operator $\mathcal{O}$. For convenience, we use an operator-state correspondence description: For any operator $\mathcal{O} = \sum_{i,j} O_{ij} |i\rangle\langle j|$, where $|i\rangle$ and $|j\rangle$ are from an orthonormal basis in the Hilbert space of consideration, we define the corresponding operator state as $|\mathcal{O}) \equiv \sum_{i,j} O_{ij} |i\rangle |j\rangle$. Since we will consider quantities at the infinite temperature, we define the inner product between two operator states $|\mathcal{A})$ and $|\mathcal{B})$ as

$$(\mathcal{A}|\mathcal{B}) \equiv \frac{\text{Tr}[\mathcal{A}^\dagger \mathcal{B}]}{\text{Tr}[I]} \, , \tag{1}$$

where $I$ is an $D \times D$ identity matrix and $D$ is the dimension of the (state) Hilbert space.

The Heisenberg evolution of an operator is

$$\mathcal{O}(t) = e^{iHt} \mathcal{O} e^{-iHt} = \sum_{n=0}^{\infty} \frac{(it)^n}{n!} \mathcal{L}^n \mathcal{O} \, , \tag{2}$$

where the Liouvillian superoperator is defined as $\mathcal{L}\mathcal{O} \equiv [H, \mathcal{O}]$. Somewhat abusing the notation, we also define $\mathcal{L}|\mathcal{O}) \equiv |[H, \mathcal{O}])$. The Heisenberg evolution can therefore be equivalently expressed as

$$|\mathcal{O}(t)) = \sum_{n=0}^{\infty} \frac{(it)^n}{n!} \mathcal{L}^n |\mathcal{O}) \, . \tag{3}$$

To motivate the Lanczos algorithm, note that we can approximate the Heisenberg evolution Eq. (3) by summing over $n$ up to $n_{\text{max}}$. The resulting approximated operator is therefore in the so-called Krylov space

$$\mathcal{H}_{\mathcal{O}} = \text{span} \{ \mathcal{L}^n |\mathcal{O}), n = 0 \ldots n_{\text{max}} \} \, . \tag{4}$$

The Lanczos algorithm is a famous routine to generate an orthonormal basis (we dub it as Krylov basis) $\{|\mathcal{O}_n)\}$, $n = 0 \ldots n_{\text{max}}$ of the Krylov space recursively, which also tridiagonalizes $\mathcal{L}$. Starting with the initial operator $|\mathcal{O}_0) \equiv |\mathcal{O})$, we have $|\mathcal{O}_1) = b_1^{-1} \mathcal{L}|\mathcal{O}_0)$, where $b_1^2 = (\mathcal{L}\mathcal{O}_0|\mathcal{L}\mathcal{O}_0)$. For $n \geq 2$, we recursively define

$$\begin{aligned} |\mathcal{A}_n) &= \mathcal{L}|\mathcal{O}_{n-1}) - b_{n-1} |\mathcal{O}_{n-2}) \, , \\ b_n^2 &\equiv (\mathcal{A}_n|\mathcal{A}_n) \, , \\ |\mathcal{O}_n) &= \frac{1}{b_n} |\mathcal{A}_n) \, , \end{aligned} \tag{5}$$

where $b_n$ is the Lanczos coefficient. In this basis, $\mathcal{L}$ becomes tridiagonal with the matrix representation

$$\mathcal{L} = \begin{pmatrix} 0 & b_1 & 0 & 0 & \ldots \\ b_1 & 0 & b_2 & 0 & \ldots \\ 0 & b_2 & 0 & b_3 & \ldots \\ 0 & 0 & b_3 & 0 & \ldots \\ \vdots & \vdots & \vdots & \vdots & \end{pmatrix} \, . \tag{6}$$

By expanding the Heisenberg-evolved operator in the Krylov basis,

$$|\mathcal{O}(t)) = \sum_{n=0}^{\infty} \varphi_n(t) |\mathcal{O}_n) \, , \tag{7}$$

the Heisenberg equation of motion becomes

$$-i\partial_t \varphi_n(t) = b_n \varphi_{n-1}(t) + b_{n+1} \varphi_{n+1}(t) \, , \tag{8}$$

for $n = 0 \dots n_{\max}$ with the initial condition $\varphi_n(0) = \delta_{n0}$, where we also define $b_0 \equiv 0$. We therefore see that, the equation of motion governing the coefficients in the Krylov basis can be viewed as a single-particle hopping problem on a semi-infinite chain, with the hopping amplitudes $b_n$ given by the Lanczos coefficients. Denoting $\vec{\varphi}(t) = (\varphi_0(t), \varphi_1(t), \dots)^T$, we have $\vec{\varphi}(t) = e^{i\mathcal{L}t}\vec{\varphi}(0)$, where $e^{i\mathcal{L}t}$ is the matrix exponential from Eq. (6).

This perspective indeed motivates a natural consideration of complexity. One intuition of the complexity of $|\mathcal{O}(t))$ comes from the complexity of $|\mathcal{O}_n)$. In particular, $|\mathcal{O}_n)$ involves an $n$-nested commutator with the Hamiltonian, and the operator is therefore more complex and more nonlocal with the increasing order of $n$. The order $n$ therefore can be served as a measure of the operator complexity, which motivates the definition of Krylov complexity in Ref. [25],

$$C_K(t) \equiv \sum_{n=0}^{\infty} n |(\mathcal{O}(t)|\mathcal{O}_n)|^2 . \tag{9}$$

It can also be interpreted as the mean position of the particle in the emergent single-particle hopping problem. On the other hand, $\vec{\varphi}(t)$ also gives us a measure of how much resource one has to use to have a good approximation of $|\mathcal{O}(t))$. If $\vec{\varphi}(t)$ is concentrated or localized at the small $n$, then one does not need too high of a truncation order $n_{\max}$ to obtain a good approximation of $|\mathcal{O}(t))$ in the Lanczos algorithm. As we will see, this is indeed the case for the operators in the MBL systems.

It is worth to mention that one would need both $\vec{\varphi}(t)$ (or equivalently $b_n$) and $|\mathcal{O}_n)$ to reconstruct the operator $|\mathcal{O}(t))$ fully. However, there are physical quantities which can be obtained from $\vec{\varphi}(t)$ or $b_n$ solely. A notable example is the auto-correlation function

$$F(t) \equiv \frac{\text{Tr}[\mathcal{O}(t)^\dagger \mathcal{O}(0)]}{\text{Tr}[I]} = (\mathcal{O}_0|\mathcal{O}(t)) = \varphi_0(t) . \tag{10}$$

The spectral function, defined by its Fourier transformation $\Phi(\omega) = \frac{1}{2\pi} \int_{-\infty}^{\infty} F(t)e^{-i\omega t} dt$, can then be expressed as

$$\Phi(\omega) = \sum_m |(\mathcal{O}_0|E_m)|^2 \delta(\omega - E_m) , \tag{11}$$

where $|E_m)$ is an eigenstate of $\mathcal{L}$ in Eq. (6) with an eigenvalue $E_m$. From the single-particle hopping problem perspective, this is the local density of states on the first site. We also note that the analytical continuation of the spectral function can also be formally expressed through the continuous fraction of $b_n$, which is obtained by solving Eq. (8) with Laplace transformation [25, 59].

## 3   Operator growth in a MBL system

In this section, we study the operator growth problem from the perspective of Lanczos algorithm in a MBL system. We consider the one-dimensional (1D) spin chain with $L$ sites and the open boundary condition

$$H = -J \sum_{\langle ij \rangle} Z_i Z_j - g \sum_j X_j + \sum_j h_j Z_j , \tag{12}$$

where $\langle ij \rangle$ denotes the nearest neighbors, $X_j, Y_j, Z_j$ are the Pauli matrices on the site $j$, $J = 1$ is the energy unit, $g = -1.05$, and $h_j$ is the random transverse field drawing from a uniform distribution $[-h, h]$. (For an odd chain, we label sites $j = -(L-1)/2 \dots (L-1)/2$; for an even chain, we label

sites $j = -L/2+1 \ldots L/2$.) Such a spin chain exhibits MBL when the disorder strength $h \gtrsim 3.5$ [52], also as shown in Appendix A from the gap-ratio statistics. For all of our following calculations, we generate $10^3$ disorder realizations, and all the disorder-averaged quantities are denoted with an overline.

## 3.1 Lanczos coefficients

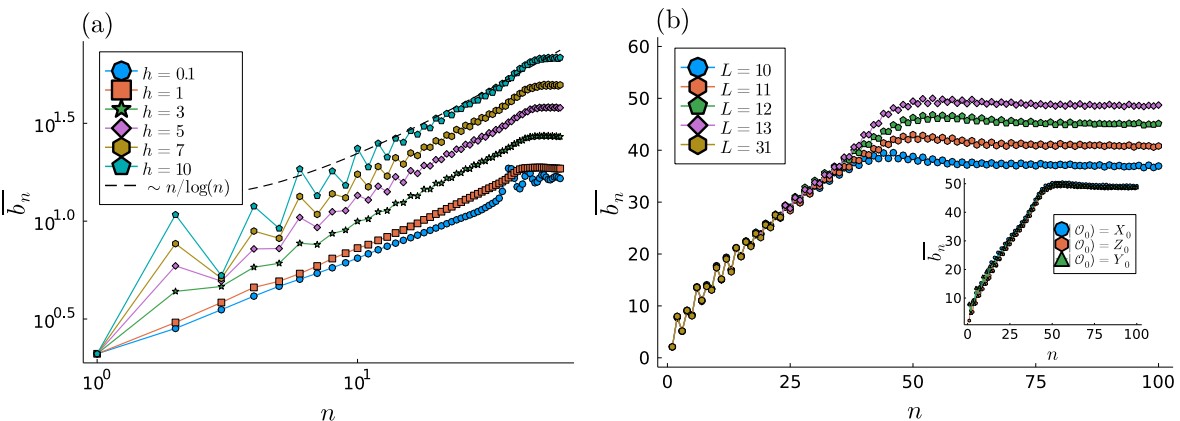

Figure 1: (a)The disorder-averaged Lanczos coefficients $\overline{b_n}$ for several disorder strengths $h$ and system size $L = 13$ in the random field Ising model, starting with the initial operator $Z_0$. In the MBL regime $h \gtrsim 3$, $\overline{b_n}$ has the same asymptotic behavior as in the ergodic regime, but with an additional even-odd alteration. (b)Disorder-averaged Lanczos coefficients $\overline{b_n}$ for several system sizes $L$ and $h = 7$, with the initial operator $Z_0$. The onset point of the plateau of $\overline{b_n}$ increases with the system size $L$. We therefore conclude that the saturation of $\overline{b_n}$ is due to the finite size. Inset: the disorder averaged Lanczos coefficients starting with the the $X_0$, $Y_0$ and $Z_0$ operators for $L = 13$ and $h = 7$. The even-odd alteration of the Lanczos coefficients is independent of the choice of the initial operators.

We start by generating the Lanczos coefficients and examining their behaviors for various disorder strengths $h$. In Fig. 1(a), we show the disorder-averaged Lanczos coefficients for the system size $L = 13$ and the initial operator $\mathcal{O}_0 = Z_{j=0}$. We note that, before saturating or plateauing at large $n$, $\overline{b_n}$ appears to have an asymptotic behavior $\sim n/\ln(n)$, which is consistent with Refs. [25, 51]. In the linear-linear plot (Fig. 1(b)), the traces indeed appear to be linear and the logarithmic correction is difficult to notice. However, as we see in the log-log plot in Fig. 1(a), the logarithmic correction has the effect of skewing the apparent exponent $\overline{b_n} \sim n^\delta$ of $\delta$ from $\delta = 1$ to $\delta \approx 0.8$. We also plot our data with $\sim n/\ln(n)$ to show that the logarithmic correction is indeed present, and such a behavior is expected in both the MBL and ergodic systems. On the other hand, we observe that in the MBL regime $h \gtrsim 3$, the disorder-averaged Lanczos coefficients $\overline{b_n}$ starts to show an even-odd alteration, with the amplitude of the alteration increases with an increasing disorder strength $h$.

In Fig. 1(b), we plot $\overline{b_n}$ for different system sizes $L$. We see that the starting $n$ of the saturation or plateau increases as the system size $L$ increases. We therefore conclude that the plateau behavior of $\overline{b_n}$ is due to the finite system size $L$. In fact, this appears to be a common finite-size feature of the Lanczos coefficients in various models [33, 59].

To further confirm that the behavior before the plateau is representative of the thermodynamic limit, we push the calculation up to $L = 31$ using a different algorithm by representing the operators $|\mathcal{O}_n)$ with the Pauli-string basis. This enables us to reach a larger system size $L$ but limits the calculation to smaller $n_{\max}$ since it requires exponentially more resources to represent $|\mathcal{O}_n)$ with an increasing $n$. We note that the disorder-averaged Lanczos coefficients for $L = 31$ still show the even-odd alteration behavior. This again supports our conclusion that the asymptotic scaling and the even-odd alteration of Lanczos coefficients before the saturation is representative in the thermodynamic limit. In the inset of Fig. 1(b), we also show that the aforementioned observed behaviors of $b_n$ is independent of the initial local operator.

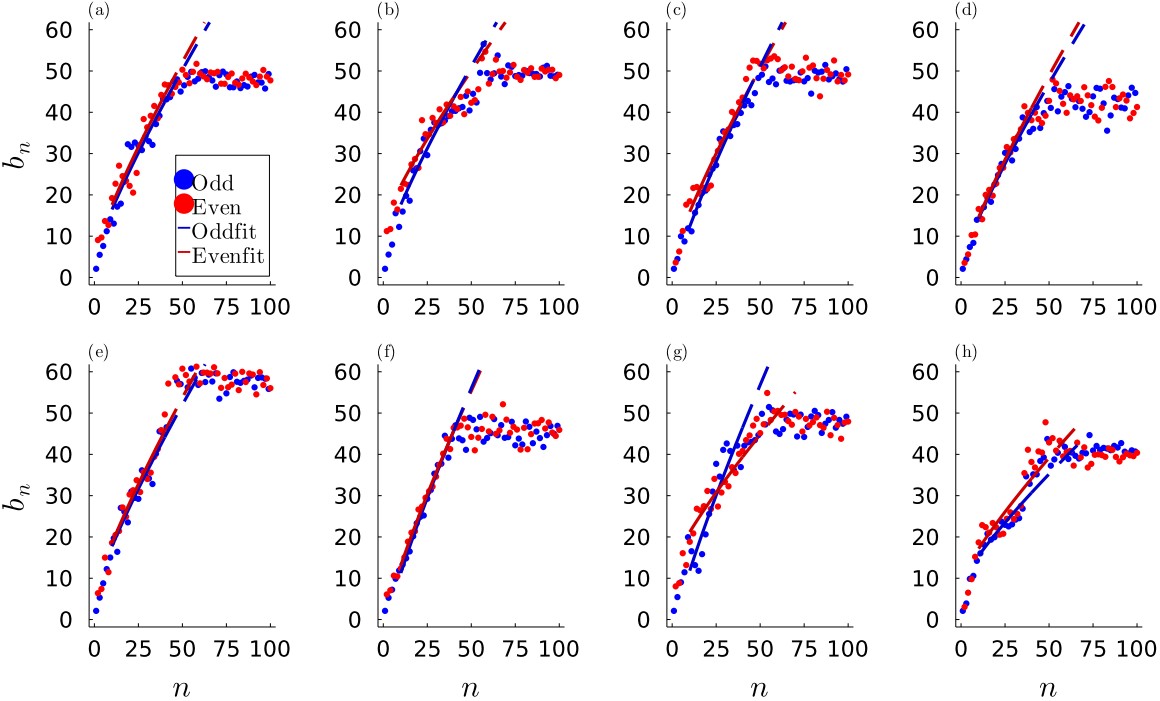

Figure 2: (a)-(h) The disorder realizations of $b_n$ for Eq. (12) for $h = 7$ and $L = 13$, with the initial operator $Z_0$. We use $b_n$ in the range of $n = 10$ to $n = 40$ to fit the linear-extrapolation formula Eq. (13), and then use the fitted results to extrapolate $b_n$ for $n > 40$ for each disorder realization.

Note that the even-odd alteration in the Lanczos coefficients has been observed in various models. For example, in Refs. [32, 33], the even-odd alteration shows up in the almost strong zero mode calculation; in Ref. [31], such an alteration also shows up in some conformal field theories; from the toy examples in Ref. [59], these even-odd alterations are often associated with the singular behaviors at the edge of the spectral function or a $\delta$-function peak at the zero frequency in the spectral function. However, a distinctive feature in our case is that such a behavior is observed after disorder averaging. In Fig. 2, we show several disorder realizations of $b_n$. (For additional data, see Appendix B.) We observe that, the even-odd alteration is hidden behind some apparent randomness in the $b_n$ sequence. This apparent randomness could result in a qualitatively different behavior of the emergent single-particle hopping problem compared to the hopping problem with clean even-odd alteration.

## 3.2 Linear extrapolation of the Lanczos coefficients

An interesting and important aspect of the recursive method or the Lanczos algorithm is that it provides us an alternative way to extrapolate the system to the thermodynamic limit by extrapolating the behavior of $b_n$. Here, we also attempt to extrapolate $b_n$ in the MBL system to the thermodynamic limit, but using the simplest linear extrapolation scheme. As mentioned in Sec. 1, it is a very challenging task to infer the thermodynamic-limit behavior from the finite-size results in the MBL systems. A much more sophisticated extrapolation scheme is therefore likely needed, such as the recipes developed in Ref. [59] or procedures accounting the possibility of the diminishing even-odd alteration [32, 33]. Again, we stress that here we only consider the simplest linear extrapolation as a first step.

From the observation of the even-odd alteration (Fig. 1) and the apparent randomness (Fig. 2), together with the asymptotic upper and lower bounds in Refs. [25,51] (despite the different choice of the disorder distribution), we extrapolate the Lanczos coefficients in the MBL system using the following linear extrapolation formula:

$$
b_n = \begin{cases} a_e \, \frac{n}{\ln n} + c_e + \Gamma_n, & \text{if } n \text{ even}, \\ a_o \, \frac{n}{\ln n} + c_o + \Gamma_n, & \text{if } n \text{ odd}. \end{cases} \tag{13}
$$

In particular, in the above formula, we expect the parameters $a_{e(o)}$ and $c_{e(o)}$ to be different for *each disorder realization*, drawn from some probability distributions. On the other hand, we also expect an additional effective randomness $\Gamma_n$ for each $n$, drawn from a probability distribution independent of $n$.

To further corroborate our extrapolation scheme, for each disorder realization, we take the Lanczos coefficients from $n = 10$ to $n = 40$ for $L = 13$ and $h = 7$, fitting them with $y_{\text{fit}} = a_{e(o)} n / \ln n + c_{e(o)}$ via the least-square fit for the even and odd branches respectively. The resulting fits are shown in Fig. 2 for some disorder realizations and the histogramw of the fitted parameters are shown in Fig. 3 (a)-(d). After fitting the data, we then examine the difference between the data and the fit $y - y_{\text{fit}}$, and we plot the histogram of the differences collected from $n \in [10, 40]$, as shown in Fig. 3(e). The histograms in Fig. 3 all look very close to Gaussian distributions, which supports our conjecture that these parameters can be viewed as drawn from some probability distributions independently. The means and the variances of the probability distributions can be estimated from the Gaussian distributions shown in Fig. 3, according to the central limit theorem.

After fitting the data using the formula Eq. (13), we extrapolate the Lanczos coefficients to large $n$ ($n > 40$) and to the thermodynamic limit as the following. For each disorder realization of $b_n$, we obtained the fitted parameters of $a_{e(o)}$ and $c_{e(o)}$ using the data in the range $n = 10$ to $n = 40$ as mentioned previously. We then take these $b_n$ from the numerical results up to $n = 40$, and then extrapolate and patch $b_n$ for $n > 40$ to $n_{\max} = 1000$ using Eq. (13) with the fitted $a_{e(o)}$ and $c_{e(o)}$ for this realization, adding the randomness $\Gamma_n$ drawn from a Gaussian ensemble with the mean and the variance given by the parameters in Fig. 3(e). In the following, we will study various quantities using the original $b_n$ and the linearly-extrapolated $b_n$.

## 3.3 Spectral functions

In this subsection, we examine the behavior of the spectral functions obtained from the original and extrapolated $b_n$ using Eq. (11). In terms of the emergent single-particle hopping problem, the spectral function is the local density of states on the first site, which we obtain by diagonalizing

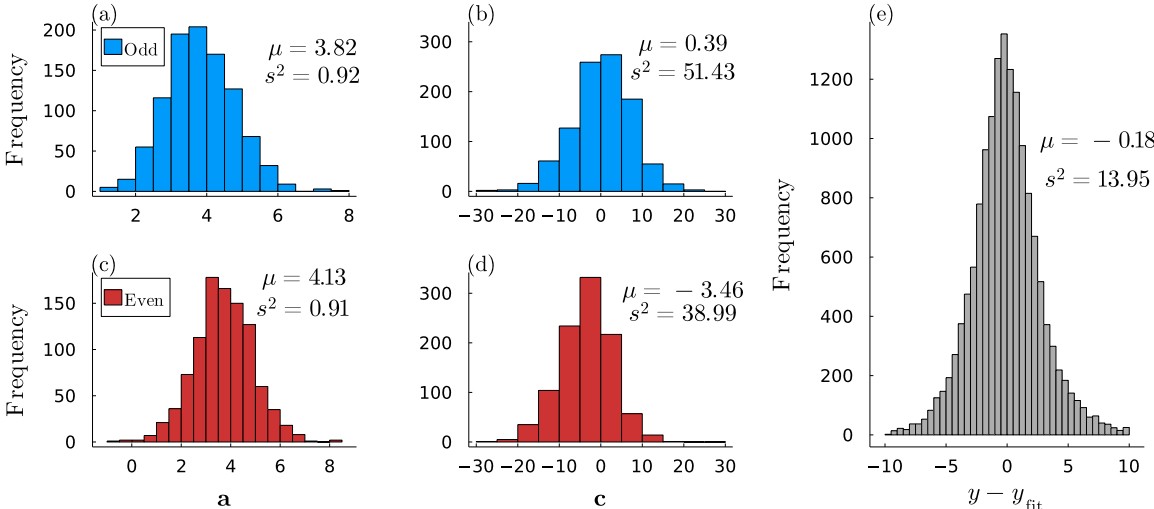

Figure 3: (a)(b) The histograms of the fitted parameters $a_o$ and $c_o$ from Eq. (13) for the odd branch. (c)(d) The histograms of the fitted data $a_e$ and $c_e$ for the even branch. (e) The histogram of $\Gamma_n$ in Eq. (13), obtained by collecting the difference between the data and the fitted results $y - y_{\text{fit}}$ for each realization and $n \in [10, 40]$. The Gaussian distributions of the histogram supports the conjectured effective randomness of the parameters. We use the $b_n$ sequence obtained from $L = 13$ and $h = 7$ in the range of $n = 10$ to $40$ for the extrapolation.

Eq. (6). We attempt to extract the analytical behaviors of the spectral function at low and high frequencies.

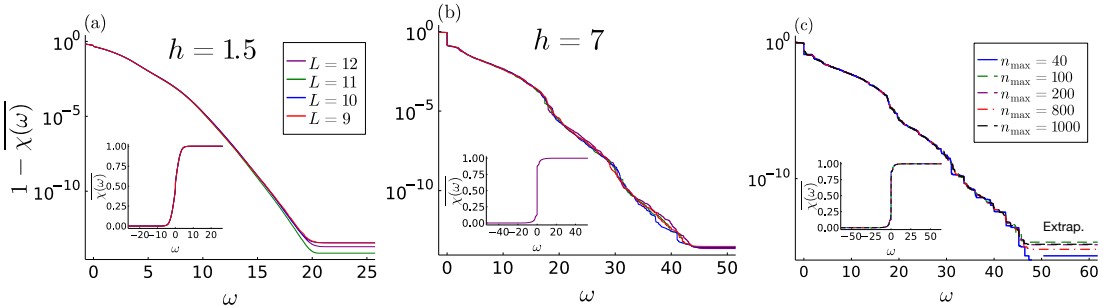

Figure 4: (a) The disorder-averaged cumulative spectral function $1 - \overline{\chi(\omega)}$ in the ergodic regime $h = 1.5$ for several system sizes $L$. (b) Same as (a) but in the MBL regime $h = 7$. (c) The disorder-averaged cumulative spectral function $1 - \overline{\chi(\omega)}$ from the extrapolated $b_n$ with several Krylov cutoff order $n_{\text{max}}$. Insets: The disorder-averaged cumulative spectral function $\overline{\chi(\omega)}$. Note that the flattening of $1 - \overline{\chi(\omega)}$ at the high frequency in all of the figures is due to the numerical round-off error.

We first examine the high-frequency behavior of $\overline{\chi(\omega)}$ from the original and the extrapolated $b_n$. Since we can only obtain discrete eigenvalues from diagonalizing Eq. (6) numerically, the spectral function will be a sum of delta functions with some weights. At the high frequency, it is

therefore more customary to consider the cumulative spectral function

$$\chi(\omega) \equiv \int_{\Omega=-\infty}^{\omega} \Phi(\Omega)d\Omega = \sum_{E_j < \omega} |(\mathcal{O}_0|E_j)|^2 , \qquad (14)$$

and extract the smooth part of the cumulative spectral function. This also gives us a convenient way to average over disorder realizations, where one would just sum over all the weights below $\omega$ in Eq. (14) from all the disorder realizations and then divide it by the number of realizations.

In Ref. [25], it is shown that if the Lanczos coefficients grow linearly in $n$, then the spectral function at the high frequency decays exponentially $\Phi(\omega) \sim \exp(-\frac{\pi|\omega|}{2a})$, where $a$ is the coefficient of the linear growth $b_n \sim an$. While it is not clear what is the effect of the plateauing of $b_n$ in the original data or the $\ln n$ correction in the extrapolated data on the spectral function, we expect that numerically, the exponential decay of the spectral function will still be a good description.

In Figs. 4(a) and (b), we plot the disorder-averaged cumulative spectral function $\overline{\chi(\omega)}$ and $1 - \overline{\chi(\omega)}$ from the original $b_n$ for $h = 1.5$ (ergodic) and $h = 7$ (MBL), several system sizes $L$ and $n_{\max} = 1000$; while we plot the same quantities in Fig. 4(c) from the linearly extrapolated $b_n$ but for several cutoff $n_{\max}$. (Note that $\overline{\chi(\omega \to \infty)} = 1$.) From the figures, we see that $\overline{\chi(\omega)}$ decays (almost) exponentially at the high frequency, and therefore so does $\overline{\Phi(\omega)}$ for both the original and extrapolated $b_n$. We extract the exponential decay in Fig. 4(c) as $\chi(\omega) \sim \exp(-\frac{\pi\omega}{2a})$, where $a \approx 4.36$ , which is close to the mean values of $a_{e(o)}$ as shown in Fig. 3. We see that the high frequency behavior of the spectral function is indeed blind to the MBL or ergodic regimes [51], where both show (close to) exponential decay.

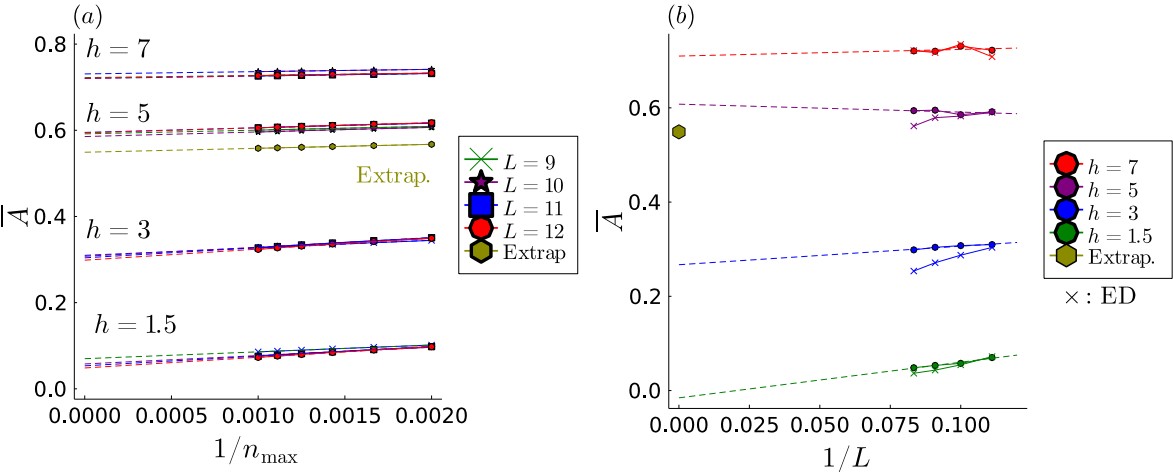

Figure 5: (a)The finite-$n_{\max}$ extrapolation of the disorder-averaged amplitude of the zero-frequency delta function for several system sizes $L$ and disorder strengths $h$. (b)The finite-size $L$ extrapolation of the disorder-averaged amplitude of the zero-frequency delta function for several disorder strengths $h$. The results are obtained from the extrapolations in (a). We also benchmark the results of the amplitudes obtained from ED.

On the other hand, at the zero frequency, we observe the presence of a delta function in the spectral function $\Phi(\omega) \sim A\delta(\omega) + \ldots$ (manifested as a step function for the cumulative spectral function $\chi(\omega)$). To infer its potential existence in the thermodynamic limit, we first extract the disorder-averaged amplitudes of the delta function $A = |(\mathcal{O}_0|E_j = 0)|^2$ at different cutoffs $n_{\max}$ for

various system sizes $L$ and disorder strengths $h$ from the original $b_n$ as shown in Fig. 5(a). We then extrapolate the amplitudes to $n_{\max} = \infty$ by linearly extrapolating it with $1/n_{\max}$. The amplitudes extrapolated at $n_{\max} = \infty$ are then plotted in Figs. 5(b), where we attempt a finite-size scaling in $1/L$. As shown in the figures, the amplitudes $A$ in MBL ($h = 7$ and $h = 5$) retain a sizeable number, while the amplitude in the ergodic system ($h = 1.5$) is extrapolated to a value close to zero. Note that we also benchmark the amplitudes obtained from the Lanczos method with the ones obtained from exact diagonalization (ED). While for $h = 7$, the amplitudes obtained from both methods appear to agree with each other pretty well, we observe that the Lanczos method tends to overestimate for other disorder strengths $h$. The existence of the delta function is indeed a hallmark of MBL, since it implies that the auto-correlation function $F(t)$ will decay to $A \neq 0$ at the infinite time, assuming the rest part of the spectral function is analytical. We also note that the extrapolated $b_n$ also results in a delta-function at the zero frequency with a sizable amplitude, though having a value relatively lower than the finite-size result ($h = 7$).

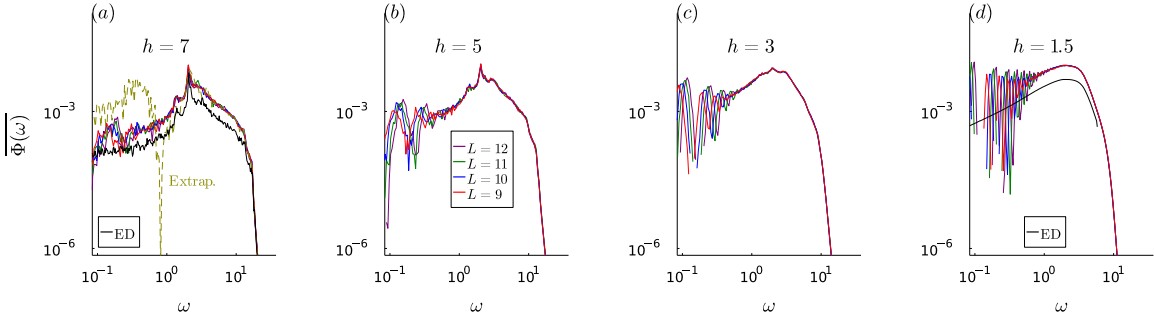

Figure 6: The low-frequency part of the disorder-averaged spectral function $\overline{\Phi(\omega)}$ for the Krylov order $n_{\max} = 1000$, several system sizes $L$ and disorder strengths (a) $h = 7$ (b) $h = 5$ (c) $h = 3$ (d) $h = 1.5$. In (a) and (d), we benchmark the results with the spectral function obtained from exact diagonalization (ED) at the system size $L = 11$. (Note that we impose a cutoff frequency in the ED result for the ease of computation.) In (a), we also plot the spectral function $\overline{\Phi(\omega)}$ obtained from the extrapolated $b_n$.

It is also interesting to see if the spectral function shows any singular behavior at the low frequency in addition to the zero-frequency delta function. The low-frequency singular behavior of the spectral function would manifest in the long-time behavior of the auto-correlation function. In Figs. 6(a)-(d), we plot the low-frequency regime of the disorder-averaged spectral function $\overline{\Phi(\omega)}$ from the original $b_n$ for the system size $L = 12$ and various disorder strengths $h$ using logarithmic binning in $\omega$. We also plot the spectral function obtained from diagonalizing $H$ (Eq. (12)) for the system size $L = 11$ and disorder strengths $h = 7$ and $h = 1.5$ in Figs. 6(a) and (d), respectively. This let us benchmark the results of the spectral functions obtained from the Lanczos algorithm with the ones obtained from ED. Additionally, we also plot the spectral function obtained from the extrapolated $b_n$ in Fig. 6(a).

We note that it is fairly challenging to extract the low-frequency behavior from our present numerical data, and that the Lanczos algorithm appears to be better at capturing the high-frequency part of the spectral function. The resolution of the low-frequency part of the spectral function is determined by the smallest positive nonzero eigenvalue from diagonalizing $\mathcal{L}$ in Eq. (6). With the cutoff Krylov order $n_{\max} = 1000$, the smallest positive nonzero eigenvalue typically has the order of $10^{-1}$, and roughly 10 eigenvalues fall in the window of $\omega \lesssim 10^0$ for each disorder realization.

This results in the artificial erratic behavior of the spectral functions at the range $\omega \lesssim 10^0$ shown in Fig. 6, especially when the disorder strength $h$ is small. Nevertheless, as we see in Figs. 6(a) and (d), the spectral functions obtained from the Lanczos method agrees with the ones obtained from ED qualitatively well.

In the figures, the spectral functions at the low frequency $\omega \ll 10^0$ appear to show a power-law behavior $\omega^\alpha$ with positive exponents $\alpha > 0$ irregardless of the disorder strengths $h$. Additionally, in Figs. 6(a)(b), we also observe that the spectral functions in the MBL regime have another range $\omega \approx 10^0$ to $\omega \approx 10^1$ showing a power-law behavior with a negative exponent, which is very close to $\alpha \approx -1$.

An interesting open question is to obtain an analytical low-frequency behavior of the spectral function from the linear extrapolation formula Eq. (13) of $b_n$ and to compare with the disorder-averaged spectral function obtained numerically [65–70]. On the other hand, if one has an expected low-frequency behavior of the spectral function, one can also use this information to modify the extrapolation procedure of $b_n$ [59,71]. In any case, we believe that the low-frequency behavior of the spectral function in the MBL systems warrants a further investigation in the future using the Lanczos method [71].

## 3.4 Local integrals of motion and zero modes

The emergent single-particle hopping problem Eq. (6) is a nearest-neighbor hopping problem on a bipartite lattice. If $n_{\max}$ is even (so the total number of lattice sites is odd), it is guaranteed to have an eigenstate with a zero eigenvalue $\mathcal{L}|E_j = 0) = 0$. The eigenstate is given by $|E_j=0) \propto \sum_{n=0}^{n_{\max}} f_n |\mathcal{O}_n)$, where $f_n = 0$ if $n$ is odd and

$$\frac{f_{2m}}{f_0} = (-1)^m \prod_{j=1}^{m} \frac{b_{2j-1}}{b_{2j}} \,, \tag{15}$$

if $n = 2m$ is even.

Since $\mathcal{L}|E_j = 0) = 0$, the operator $|E_j = 0)$ is an integral of motion of the original many-body dynamics. From the emergent single-particle perspective, if $|E_j = 0)$ is localized around $n = 0$, then $|E_j=0)$ is a good candidate for the local integral of motion since the range of the support of $|\mathcal{O}_n)$ on the original spin system increases with $n$. Furthermore, such a local integral of motion will have a large overlap with the initial operator $|\mathcal{O}_0)$. The existence of the local integrals of motion is another defining feature of MBL [60–64].

In Fig. 7, we show $\overline{|(\mathcal{O}_n|E_j=0)|^2}$, the disorder-averaged wavefunction profile of the zero mode for different disorder strengths $h$, several different system sizes $L$ and $n_{\max} = 1000$. The profile of the zero modes appears to have two exponential decays: The first faster exponential decay happens at small $n$ and transitions into the second slower exponential decay at around $n = 40$ to $n = 50$, which is also the range of $n$ that $b_n$ plateaus. Indeed, one can also see from the insets of Fig. 7 that the range of the first faster exponential decay extends when the system size $L$ increases. We therefore conclude that the first faster exponential decay is likely a more representative behavior in the thermodynamic limit, while the second exponential decay is likely due to the plateauing behavior of $b_n$, which is a finite-size effect. Curiously, it appears that the zero modes in both MBL regime and ergodic regime show such a behavior, though the localization of the zero mode at the small $n$ should be expected from Eq. (15); The (almost) linearly growing $b_n$ at the small $n$ indeed gives $b_{2n-1}/b_{2n} < 1$, which results in the localization of the zero mode at the small $n$. The qualitative difference of the zero mode and the potential integral of motion between the MBL

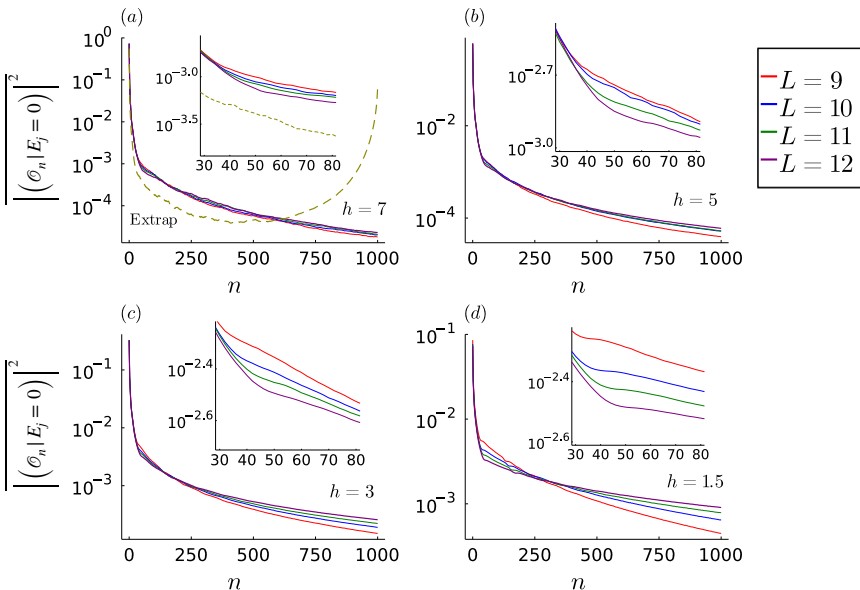

Figure 7: The disorder-averaged wavefunction profile of the zero mode obtained from the $b_n$ for the Krylov order $n_{\max} = 1000$, several system sizes $L$ and disorder strengths (a) $h = 7$, (b) $h = 5$, (c) $h = 3$ and (d) $h = 1.5$. Insets: same data as in the main figures but zoomed-in in the range of $n = 30\text{-}80$. In (a), we also show the disorder-averaged zero mode obtained from the extrapolated $b_n$.

regime and the ergodic regime may lie in a more detailed scaling behavior of the exponential localization length versus the system size or the content of $|\mathcal{O}_n\rangle$. Analysing the zero modes and connecting them to the local integrals of motion in the MBL regime are therefore a worthy direction to explore, which we leave for future work.

In addition, we also show the behavior of $\overline{|(\mathcal{O}_n|E_j=0)|^2}$ for the extrapolated $b_n$ in Fig. 7(a). We observe, however, that the zero mode is localized not only on the edge of $n = 0$ but also on the edge of $n = n_{\max}$. This behavior is reminiscent of the results in the almost-strong edge mode in Ref. [32, 33]. In this case, one can attempt to construct an *approximate* integral of motion by truncating $|E_j=0\rangle$ at some $n = n^*$, or $\Psi \propto \sum_{n=0}^{n^*} f_n \mathcal{O}_n$, such that $[H, \Psi]$ is minimized under some norm and some proper normalization of $\Psi$. However, we also note that a subtle order of limits has to be considered. That is, the proper way of taking the thermodynamic limit is to take $n_{\max} \to \infty$ first and then take $L \to \infty$. On the other hand, the zero mode result from the extrapolated $b_n$ in Fig. 7(a) is in a sense considering the limit $L \to \infty$ first at some finite $n_{\max}$. The results from the non-extrapolated $b_n$ in Fig. 7(a) suggests that the localization of the zero mode at the edge $n_{\max}$ might be an artifact from the linear extrapolation or from the different orders of the limits.

The effect of $b_n$ even-odd alteration on the zero mode is also clear from Eq. (15). In particular, for a clean system, if the ratio $b_{2n-1}/b_{2n} < 1$ for almost all $n$ after some $n = n^*$, then the zero mode will localize around edge $n = 0$. However, if $b_{2n-1}/b_{2n} < 1$ but $b_{2n-1}/b_{2n} > 1$ after some $n = n^*$, then the zero mode will localize around both edges. Since there is an apparent randomness in $b_n$ in our problem, Eq. (15) suggests an alternative way to examine the statistics of $b_n$, which is to examine the statistics of $b_{2n-1}/b_{2n}$ or $\ln(b_{2n-1}/b_{2n})$ from different disorder realizations. In Figs. 17 and 18 (see Appendix B), we show the histograms of $\ln(b_{2n-1}/b_{2n})$ for several $n$, system size $L = 12$, and disorder strengths $h = 1.5$ and $h = 7$. We observe that $\ln(b_{2n-1}/b_{2n})$ appears

to have a Gaussian distribution statistically with a negative mean for most of the $n$. We therefore concludes that the "typical" zero mode is likely to be localized around $n = 0$.

### 3.5   Krylov complexity, wavefunction profile and return probability

As mentioned in Sec. 2, we can view the operator growth problem as a single-particle hopping problem on a semi-infinite chain. The time evolution of the single particle wavefunction $\vec{\varphi}(t)$ reflects the growth of the operator in the Krylov basis $|\mathcal{O}(t)) = \sum_n \varphi_n(t)|\mathcal{O}_n)$. One can therefore examine various properties of $\vec{\varphi}(t)$ to characterize the operator growth.

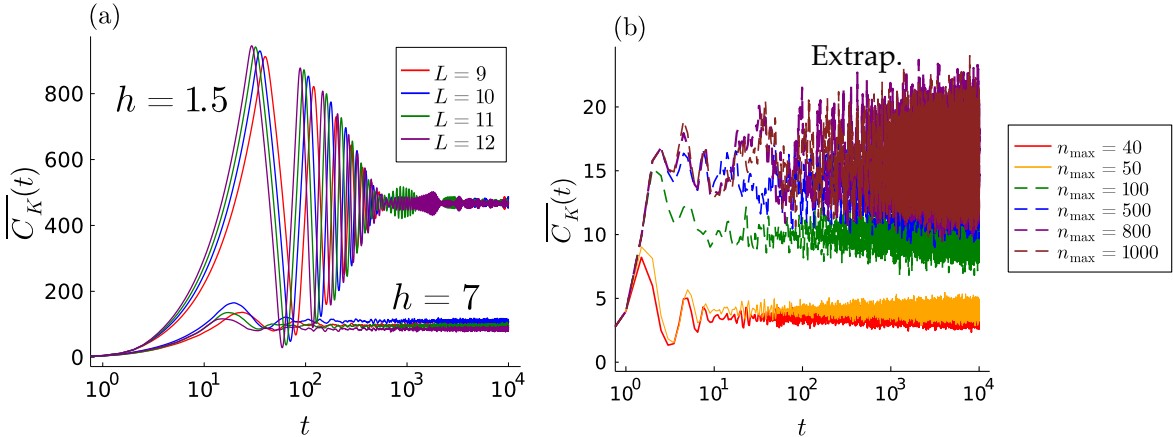

Figure 8: (a) The disorder-averaged Krylov complexity $\overline{C_K}(t)$ for the Krylov order $n_{\max} = 1000$, several system sizes $L$, and the disorder strengths $h = 1.5$ (ergodic regime) and $h = 7$ (MBL regime). Note that at a very short time, $\overline{C_K}(t)$ in the ergodic regime reaches $n_{\max} = 1000$ already, while $\overline{C_K}(t)$ stays small relative to $n_{\max} = 1000$ at all times in the MBL regime. (b) Disorder-averaged Krylov complexity $C_K(t)$ from the extrapolated Lanczos coefficients with several Krylov truncation order $n_{\max}$. Note the scale difference of y-axis compared to (a).

First, we examine the mean position in time of the emergent single-particle wavefunction, namely the Krylov complexity defined in Eq. (9), from both the original and the extrapolated $b_n$. In Fig. 8(a), we show the disorder-averaged Krylov complexity in the ergodic regime ($h = 1.5$) and the MBL regime ($h = 7.0$) for various system sizes $L$, while we show the result from the extrapolated $b_n$ with several $n_{\max}$ in Fig. 8(b). In the ergodic regime, the Krylov complexity indeed grows as a stretched exponential ($\sim e^{\sqrt{At}}$) at short times before it reaches the Krylov cutoff order $n_{\max} = 1000$, consistent with Ref. [25]. Here we stress that, ideally, one should consider the limit $n_{\max} \to \infty$ first before considering the long-time limit. As we see from Fig. 8(a), in the ergodic regime, the "particle" reaches the Krylov cutoff order $n = n_{\max}$ exponentially fast. We therefore expect that various results at the long time in the ergodic regime will have much severe finite-$n_{\max}$ effect, and one should be aware of this potential issue when interpreting the numerical results. On the other hand, in the MBL regime, we see that the Krylov complexity is relatively low throughout compared to the truncated Lanczos order $n_{\max} = 1000$. The long-time averaged Krylov complexity for the MBL systems appears to be bounded in time and saturates quickly to a value much smaller than the Krylov truncation order $n_{\max}$ from both extrapolated and original $b_n$.

Note that for the result from extrapolated $b_n$ in Fig. 8(b), $\overline{C_K}(t)$ shows a fluctuation around

the long-time averaged value at long times, and the magnitude of the fluctuation appears to grow logarithmically in time. However, by examining $\overline{C_K}(t)$ for several $n_{\max}$, we notice the onset time of such a fluctuation increases with $n_{\max}$, suggesting it as a result of finite $n_{\max}$. From the results in Fig. 8(a) and (b), we conclude that the Krylov complexity is bounded in time in the MBL systems.

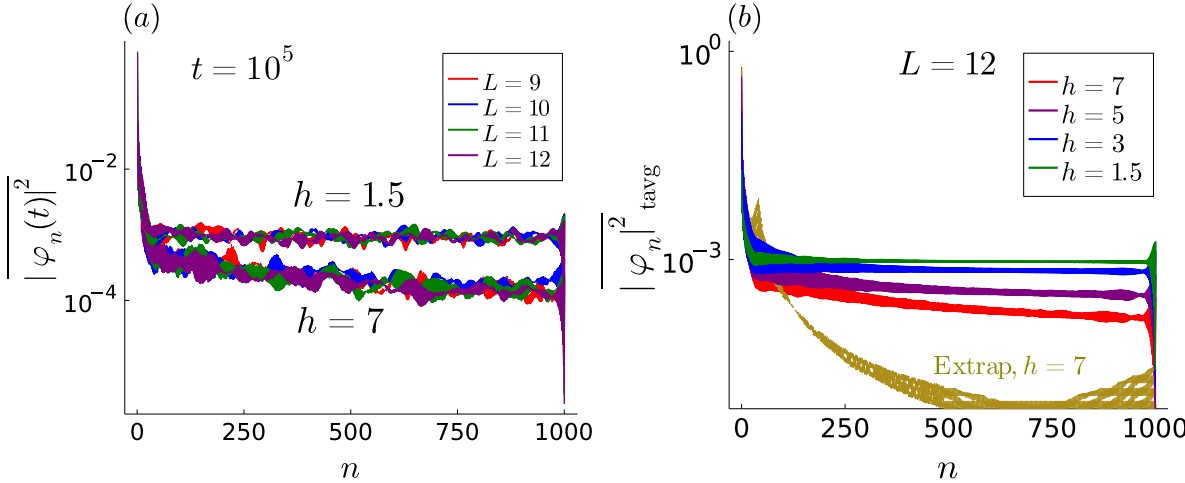

Figure 9: (a)The disorder-averaged wavefunction profile $\overline{|\varphi_n(t)|^2}$ at $t = 10^5$ in the ergodic regime ($h = 1.5$) and in the MBL regime ($h = 1.5$) for several system sizes $L$ and $n_{\max} = 1000$. (b) The long-time and disorder-averaged wavefunction profile $\overline{|\varphi_n|^2_{\text{tavg}}}$ for $L = 12$ and several disorder strengths $h$. We also show the result of $\overline{|\varphi_n|^2_{\text{tavg}}}$ from the extrapolated $b_n$.

The growth of the operator can be more easily visualized by examining the wavefunction profile, or equivalently the probability distribution of the single-particle wavefunction at various times. In Fig. 9(a), we first examine it at an exponentially long time $t = 10^5$ in the ergodic ($h = 1.5$) and MBL regime ($h = 7$) for various system sizes. We see that the wavefunctions appear to be localized around $n = 0$ for both ergodic and MBL regimes at small $n$. However, while the wavefunction profile for $h = 7$ turns into a slower exponential decay for larger $n$, the wavefunction profile for $h = 1.5$ turns into a constant for larger $n$. In Fig. 9(b), we examine the long-time averaged wavefunction profile

$$|\varphi_n|^2_{\text{tavg}} \equiv \lim_{T \to \infty} \frac{1}{T} \int_0^T |\varphi_n(t)|^2 = \sum_j |\langle \mathcal{O}_n | E_j \rangle|^2 |\langle E_j | \mathcal{O}_0 \rangle|^2 , \tag{16}$$

assuming no degeneracy. We again see that the long-time and disorder averaged wavefunctions are all localized around $n = 0$ at the small $n$, but turn into a constant or a slower exponential decay at large $n$ for the ergodic regime or MBL regime, respectively. In addition, we also show the result from the extrapolated $b_n$ in Fig. 9(b), which shows a decay at the small $n$ but increases at the large $n$. However, the majority of the weight of the wavefunction is still concentrated around $n = 0$. Again, we remind the readers about the potential issue of the finite-$n_{\max}$ effect when viewing the result in the ergodic regime at this exponentially long time.

In Fig. 10(a) and (b), we further show $\overline{|\varphi_n(t)|^2}$ in the MBL regime from the original and extrapolated Lanczos coefficients for several times $t$, respectively. We again see that the wavefunctions

are localized around $n = 0$ and hardly moving from $t = 10$ to $t = 10^5$. We therefore conclude that the emergent single-particle hopping problem in the MBL regime is localized if initialized on the first site . This is in stark contrast with the case in the ergodic phase, where the single particle wavefunction propagates to large $n$ superpolynomially fast [25].

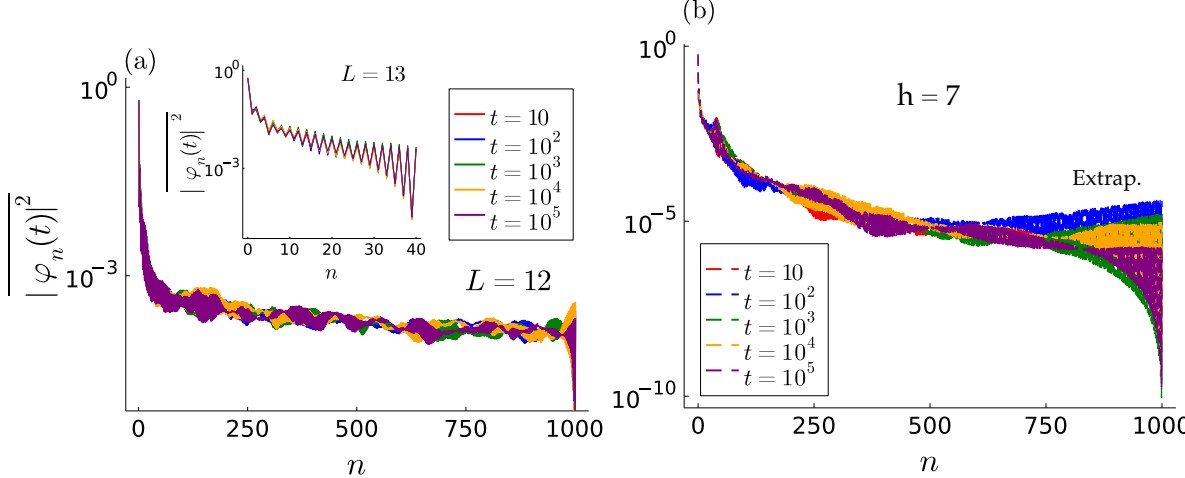

Figure 10: (a)The disorder-averaged wavefunction profile $\overline{|\varphi_n(t)|^2}$ in the MBL regime ($h = 7$) for $L = 12$ and $n_{\max} = 1000$ at several different times $t$. Inset: $\overline{|\varphi_n(t)|^2}$ in the MBL regime $h = 7$ for $L = 13$ and $n_{\max} = 40$. (b)$\overline{|\varphi_n(t)|^2}$ from the extrapolated $b_n$ for $n_{\max} = 1000$ at several different times $t$.

Our result also implies that the Krylov method can be an efficient method to simulate the operator dynamics in MBL. In practice, we have to choose a truncation $n_{\max}$ as the cut-off order of the Lanczos algorithm. If $\varphi_n(t)$ is localized at small $n$, then the truncation error $\epsilon(t) = \sum_{n=n_{\max}+1}^{\infty} |\varphi_n(t)|^2$ will be small. In particular, from our results of $|\varphi_n(t)|^2$, we expect this error will be small for MBL systems even at exponentially long times. Note that while $|\varphi_n(t)|^2$ seems to be localized and not changing too much with times, the phases of the wavefunction are still evolving. This can still cause the operator entanglement entropy or out-of-time-ordered correlator to change at long times in the MBL systems.

Finally, to further support our claim that the single-particle problem is localized when initialized on the first site, we examine the long-time and disorder averaged return probability (RP). The return probability at a time $t$ is $P(t) = |\langle \mathcal{O}_0 | e^{i\mathcal{L}t} | \mathcal{O}_0 \rangle|^2$. We therefore have the long-time averaged return probability

$$\text{RP} \equiv \lim_{T \to \infty} \frac{1}{T} \int_0^T P(t)dt = \sum_j |\langle \mathcal{O}_0 | E_j \rangle|^4, \tag{17}$$

assuming no degeneracy. First we attempt to extrapolate RP in the limit $n_{\max} \to \infty$. In Fig. 11(a), we calculate the RP for various system sizes $L$, disorder strengths $h$, and Krylov order $n_{\max}$, plotting them with $1/n_{\max}$. From the figure, we see the RPs show linear behavior with $1/n_{\max}$. Therefore, we use the simplest linear extrapolation to extrapolate them to the $n_{\max} \to \infty$ limit. In Fig. 11(b), we then attempt to extrapolate RP to the thermodynamic limit $L \to \infty$ using the simplest linear (in $1/L$) extrapolation again . We observe that the RP retains a sizable value for the MBL systems, while the RP for the ergodic system is being extrapolated to a value close to zero. The result of RP

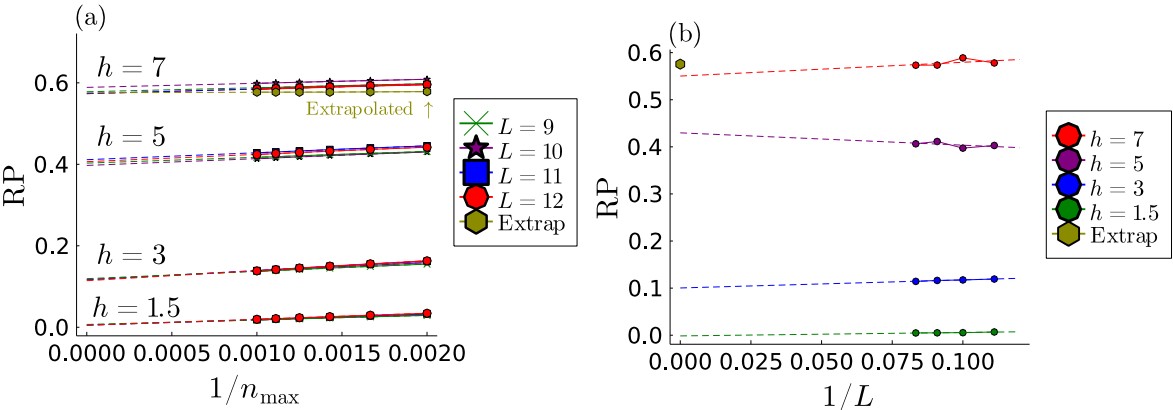

Figure 11: (a)The finite-$n_{\max}$ extrapolation of the long-time and disorder averaged return probability for several system sizes $L$ and disorder strengths $h$. (b)The finite-size $L$ extrapolation of the long-time and disorder average return probability for disorder strengths $h$. The results are obtained from the extrapolations in (a). Note that we also plot the result obtained from the extrapolated $b_n$ as a comparison.

provides another evidence of the localization of the emergent single-particle particle in the MBL regime. Again, we remind the readers that the ideal limit should be $n_{\max} \to \infty$ first and then $T \to \infty$ after, and this could affect the results in the ergodic regime more severely than in the MBL regime.

## 4 Operator growth in the MBL phenomenological model

In this section, we study the operator growth and the Krylov complexity in the MBL phenomenological model to compare with the results in the previous sections. A defining feature of the MBL system is the existence of the local integrals of motion [60–64], namely the "$\ell$-bits", which allow us to describe the MBL system by the following phenomenological model

$$H_{\text{ph}} = \sum_j J^{(0)}_j \hat{\tau}^z_j + \sum_{i<j} J^{(1)}_{ij} \hat{\tau}^z_i \hat{\tau}^z_j + \sum_{n=2}^{j-i} \sum_{i<j,\{k\}} J^{(n)}_{i\{k\}j} \hat{\tau}^z_i \hat{\tau}^z_{k_1} \ldots \hat{\tau}^z_{k_{n-1}} \hat{\tau}^z_j , \tag{18}$$

where $\{k\}$ denotes the set of sites $i < k_1, \ldots, k_{n-1} < j$, $\hat{\tau}^z_j$ is the $\ell$-bit which can be related to the physical bit $Z_j$ by a quasilocal unitary $\hat{\tau}^\alpha_j = U\sigma^\alpha_j U^\dagger$, ($\alpha = x, y, z$), and $J^{(1)}_{ij}$ and $J^{(n)}_{i\{k\}j}$ are the interactions among the $\ell$-bits which decay exponentially in the maximum distance among the set of $\ell$-bits involved in the interactions. Note that in principle, one can start from a microscopic MBL model such as Eq. (12) and construct the phenomenological model, where many subtle physics in MBL such as the many-body resonances and level repulsions will be encoded in the correlations of the $\ell$-bit interactions $J^{(n)}_{i\{k\}j}$ [68, 72]. Here, to simplify the problem, we assume $J^{(n)}_{i\{k\}j} = K^{(n)}_{i\{k\}j} \exp(-r/\xi)$ (including $n = 0$ and 1), where the range $r = j-i$, $\xi$ is $=$ the localization length, and the parameters $K^{(n)}_{i\{k\}j}$ are drawing from a uniform distribution $[-W, W]$ [22, 48]. In particular, we consider the parameters $L = 14$ as the system size, $\xi = 0.6$ and $W = 5.0$, with $10^3$ disorder realizations.

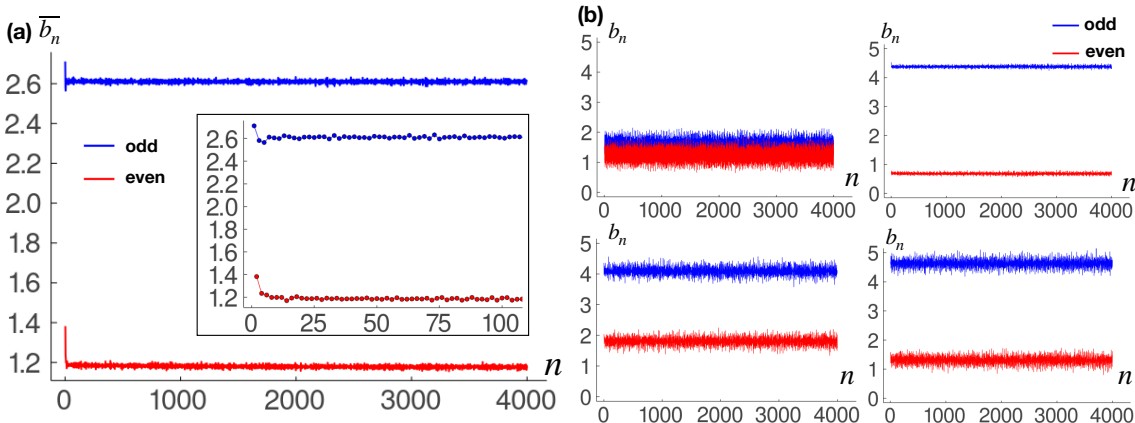

Figure 12: (a) The disorder-averaged Lanczos coefficients $\overline{b_n}$ from the MBL phenomenological model Eq. (18) for the initial operator $\hat{\tau}_0^x$ and the parameters $L = 14$, $\xi = 0.6$ and $W = 5.0$ . The Lanczos coefficients have an even-odd alteration but larger odd values, approaching constants in large $n$. Inset: The disorder averaged Lanczos coefficients $\overline{b_n}$ at small $n$. (b) The Lanczos coefficients $b_n$ from the MBL phenomenological model for several disorder realizations.

## 4.1 Lanczos coefficients

We generate the Lanczos coefficients from the Hamiltonian Eq. (18) with the initial operator $\mathcal{O}_0 = \hat{\tau}_0^x$. To calculate the Lanczos coefficients, it is easy to split the Hamiltonian into a part that involves $\tau_0^z$ and the rest: $H_{\text{ph}} = \hat{J}_{\text{eff}}\hat{\tau}_0^z + H_{\text{other}}$, where

$$\hat{J}_{\text{eff}} = J_0^{(0)} + \sum_{j \neq 0} J_{0j}^{(1)}\hat{\tau}_j^z + \sum_{n=2}^{L-1} \sum_{\{j\} \neq 0} J_{0\{j\}}^{(n)}\hat{\tau}_{j_1}^z \dots \hat{\tau}_{j_n}^z . \tag{19}$$

It is easy to see that $\hat{J}_{\text{eff}}\hat{\tau}^z$ is the only part that generates the Heisenberg evolution of $\hat{\tau}_0^x$. In fact, one can obtain $\mathcal{L}^{2n-1}\hat{\tau}_0^x = i\hat{\tau}_0^y(\hat{J}_{\text{eff}})^{2n-1}$ and $\mathcal{L}^{2n}\hat{\tau}_0^x = \hat{\tau}_0^x(\hat{J}_{\text{eff}})^{2n}$. By expressing the operator $\hat{J}_{\text{eff}}$ as a diagonal matrix of the dimensions $2^{L-1} \times 2^{L-1}$, whose diagonal entries correspond to different $\hat{\tau}^z$ configurations, we obtain the following recursive method in generating the Lanczos coefficients. Define $\hat{B}_0 = I$ as a $2^{L-1} \times 2^{L-1}$ identity matrix, $\hat{B}_1 = \hat{J}_{\text{eff}}$, $b_0 = 1$ and $b_1^2 = \frac{1}{2^L}\text{Tr}[\hat{B}_1^2]$, then

$$\hat{B}_{n+1} = \frac{1}{b_n}\hat{J}_{\text{eff}}\hat{B}_n - \frac{b_n}{b_{n-1}}\hat{B}_{n-1} ,$$

$$b_{n+1}^2 = \frac{1}{2^L}\text{Tr}[\hat{B}_{n+1}^2] , \tag{20}$$

for $n > 1$. This gives us $O_{2n-1} = b_{2n-1}^{-1}i\hat{\tau}_0^y\hat{B}_{2n-1}$ and $O_{2n} = b_{2n}^{-1}\hat{\tau}_0^x\hat{B}_{2n}$ from the Lanczos algorithm.

In Fig. 12(a), we show the disorder-averaged Lanczos coefficients $\overline{b_n}$ from the phenomenological model. Note that $\overline{b_n}$ appears to approach constants at large $n$, also having an even-odd alteration but with a higher-valued odd branch. We also show several disorder realizations of $b_n$ in Fig. 12(b).

We point out that the apparent different behavior of $b_n$ in the MBL phenomenological model compared to the microscopic MBL model is not surprising. The calculation and the result in the

phenomenological model is not only from a different choice of the initial operator, but also from a different choice of the operator basis, resulting in a different $\mathcal{L}$ . The behavior of $b_n$ from the MBL phenomenological model is also reminiscent of the single-spin dynamics evolving under a magnetic field $H = J_{\text{eff}}\sigma^z$ with the initial operator $\sigma^x$. However, the difference lies in the fact that in the MBL phenomenological model, $\hat{J}_{\text{eff}}$ depends on the other $\ell$-bit configurations, or equivalently, $\hat{J}_{\text{eff}}$ is a diagonal matrix or an operator instead of a number. Physically, this $\ell$-bit-dependent effective field $\hat{J}_{\text{eff}}$ will cause the decoherence of the spin.

## 4.2 Krylov complexity and wavefunction profile

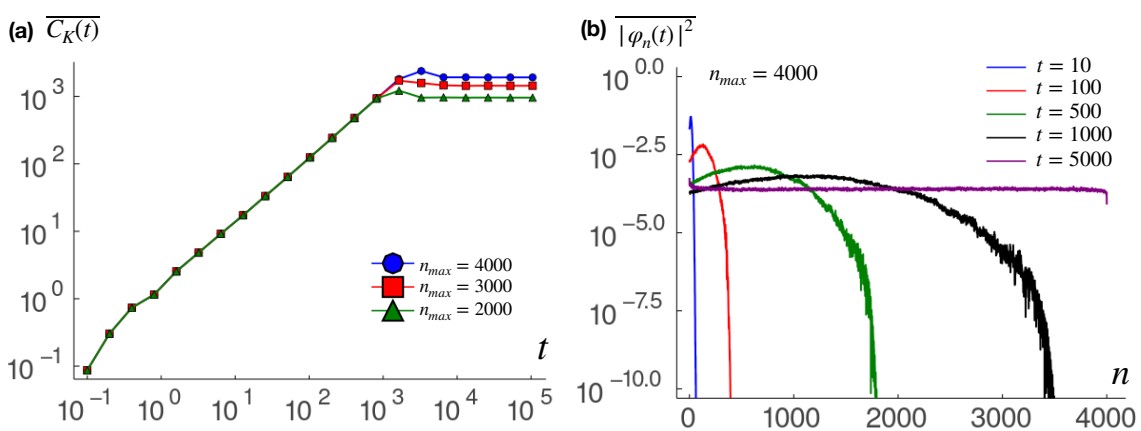

Figure 13: (a)The disorder-averaged Krylov complexity $\overline{C_K(t)}$ in the MBL phenomenological model for $L = 14$, $\xi = 0.6$ and $W = 5.0$. After a short time, $\overline{C_K(t)}$ grows linearly in time before it reaches the saturation. By comparing the results with several $n_{\text{max}}$, we conclude that the saturation is a finite-$n_{\text{max}}$ effect. (b) The disorder-averaged wavefunction profile $\overline{|\varphi_n(t)|^2}$ at different times. The wavefunction propagates towards higher $n$ linearly in time.

For each disorder realization of $b_n$, we can solve for the corresponding single-particle dynamics, representing its operator growth. In Fig. 13(a), we plot the disorder-averaged Krylov complexity $\overline{C_K(t)}$ of MBL phenomenological model. We notice that the Krylov complexity grows linearly in time after $t \gtrsim 1$ and then saturates. However, by plotting $\overline{C_K(t)}$ with several different $n_{\text{max}}$, we conclude that the saturation is due to the finiteness of $n_{\text{max}}$. Therefore, we expect the linearly growing behavior of $\overline{C_K(t)}$ will continue in the true $n_{\text{max}} \rightarrow \infty$ limit.

We also visualize the operator growth in the Krylov basis by examining the wavefunction profile $\overline{|\varphi_n(t)|^2}$ at several different times in Fig. 13(b). The wavefunction has a feature of a propagating "wavefront" towards large $n$, which appears to propagate linearly in time. The time scale when the "wavefront" hits $n_{\text{max}}$ is also the time scale when $\overline{C_K(t)}$ starts to saturate. It is therefore again clear that the saturation of $\overline{C_K(t)}$ is a finite-$n_{\text{max}}$ effect, and the single-particle wavefunction $\vec{\varphi}(t)$ will propagate towards large $n$ indefinitely if we take the $n_{\text{max}} \rightarrow \infty$ limit first. This is in contrast with the growth of the "p-bit" shown in Sec. 3, where the operator appears to be localized. Again, physically, the growth of the $\ell$-bit is due to the fact that the $\ell$-bit $\hat{\tau}_0^x$ undergoes a decoherence dynamics caused by the environmental-dependent effective magnetic field $\hat{J}_{\text{eff}}$.

## 5 Conclusion

In this work, we study the operator growth in a MBL system using Krylov basis. From this point of view, the operator growth problem can be mapped to a single-particle hopping problem on a semi-infinite chain with the hopping amplitudes given by the Lanczos coefficients. In particular, we consider the problem for the initial operators as a "p-bit" ($Z_0$) growing under a microscopic MBL Hamiltonian and as an "$\ell$-bit" ($\tau_0^x$) growing under a MBL phenomenological Hamiltonian. We find the asymptotic behavior of the Lanczos coefficients for the "p-bit" behaves as $b_n \sim n/\log(n)$ in the MBL regime, same as in the ergodic regime. However, the presence of the even-odd alteration and a relatively strong effective randomness in the MBL systems are the distinguishable features from the ergodic systems. These features could potentially be used as an alternative diagnostic for MBL. It is worth mentioning that we also examine the Lanczos coefficients in the MBL system realized by a random field Heisenberg chain (shown elsewhere [73]). We find that the aforementioned features of the Lanczos coefficients are still present, suggesting them being generic in the MBL systems and independent of the choices of the model. While it is a challenging task to extrapolate the behavior of the MBL systems from the finite sizes to the thermodynamic limit, we use the simple linear extrapolation of $b_n$ as an attempt to achieve this goal. However, we note that a more sophisticated extrapolation scheme may be required. For the "$\ell$-bit", the Lanczos coefficients also show an even-odd alteration but with a larger-valued odd branch, approaching to constants at large $n$.

With the original and the extrapolated Lanczos coefficients from the p-bit operator growth, we study various quantities related to the emergent single-particle hopping problem, including the spectral function, the zero mode, the Krylov complexity, wavefunction profile and the the return probability. We find that the spectral function decays exponentially at high frequencies for both the original and the extrapolated $b_n$ and in both the MBL and ergodic regimes. The low-frequency behavior of the spectral function is more challenging to extract, due to the limitation of the Lanczos method. However, we observe a power-law behavior with positive exponents at the low frequencies for systems in both MBL and ergodic regimes. For the systems in the MBL regime, we also observe a frequency range where the spectral function appears to have a power-law behavior with an exponent close to $-1$. We note that the spectral functions obtained from the extrapolated $b_n$ reproduces the results from the original $b_n$ very well at the intermediate to high frequencies. We also find the existence of the delta function at the zero frequency, which is a hallmark of the MBL regime. An open question is if the linearly-extrapolated $b_n$ can provide us some analytical understanding of the low-frequency asymptotic behavior of the spectral function.

The zero mode of the single-particle hopping problem is an integral of the motion of the dynamics. A localized zero mode is therefore a natural candidate as a local integral of motion. We indeed observe a localized zero mode in the MBL regime, though we also observe an apparent localization of the zero mode in the ergodic regime. A more detailed finite-size scaling of the localization length may be warranted to further distinguish the behavior of the zero modes between these two regimes.

Perhaps somewhat surprisingly, even though the asymptotic behaviour of the Lanczos coefficients are the same in both MBL and ergodic regimes, the emergent single-particle hopping problem is localized in the MBL regime if initialized on the first site. This is supported by our results of the bounded Krylov complexity in time, wavefunction profile and return probability, for both the original and the extrapolated $b_n$. This is in sharp contrast with the Krylov complexity in the ergodic regime, which is expected to grow superpolynomially in time [25], corresponding to the

delocalization of the emergent single particle. On the other hand, the Krylov complexity for "$\ell$-bit" grows linearly in time, due to the different choices of the Krylov basis. Physically, this reflects the decoherence dynamics of the $\ell$-bit $\hat{\tau}_0^x$.

We comment on some questions and future directions motivated by our results. As our result suggests, the single-particle hopping problem is localized when initialized on the first site in MBL regime but propagates superpolynomially fast and delocalizes in the ergodic regime. Since the Lanczos method provides an alternative extrapolation scheme to the thermodynamic limit, can we infer the behavior of the MBL system in the thermodynamic limit from this perspective? If so, can we study or understand the MBL-ergodic transition from this emergent single-particle localization-delocalization transition? It would also be interesting to understand the properties of the spectral function or auto-correlation function at the MBL-ergodic crossover or transition from this point of view, and to deepen our understanding of the possible experimental signatures.

Since the operator in the MBL regime is exponentially localized in the Krylov space even at the exponentially long times, Lanczos algorithm might be an efficient method to calculate the operator dynamics in the MBL systems. In particular, it means that one does not need too high of a truncation order $n_{\max}$ to obtain a good approximation of $|\mathcal{O}(t))$. However, we note that even in the MBL system, it can still require exponentially more resources to represent $|\mathcal{O}_n)$ with an increasing $n$. Can we design an efficient Krylov-based numerical method to simulate the MBL dynamics by expressing $|\mathcal{O}_n)$ more efficiently? If a Krylov-based hybrid numerical methods can achieve this, it can enable us to simulate MBL dynamics in larger system sizes and to exponentially longer times on a classical computer.

Finally, our result suggests a potential correspondence of the "dynamical regimes/phases" and "computational complexity". That is to say, the complexity of using Krylov method to calculate the operator dynamics is low for systems in the MBL regime, independent of the microscopic model. It is therefore interesting to see how much of this correspondence holds for different types of the dynamical regime or phase and how intrinsic it is. In the past few years, there have been several intriguing developments associating some notions of the complexity with the equilibrium phases of matter. For example, one hallmark of the intrinsic or symmetry protected topological phase is characterized by their circuit/state complexity [74, 75]. Namely, they cannot be converted into a tensor product state via a (symmetry preserving) finite-depth quantum circuit, while a trivial state can. Recently, there are also works suggesting that the complexity of the quantum Monte Carlo simulation (namely the existence of the sign problem) are intrinsic and associated with the types of the topological phase [76, 77]. Analogous to the above developments in using complexity to understand and characterize different phases of matter, our results could suggest a new fruitful direction in using complexity to understand and characterize dynamical regimes or phases.

*Note added*: Recently, Ref. [78] study the Krylov complexity in an integrable model, observing the suppression of Krylov complexity and the localization of the emergent single-particle problem as well.

# Acknowledgements

We thank Timothy Hsieh, Dario Rosa, Ruth Shir, and Liujun Zou for the valuable and enlightening discussions. We acknowledge supports from Perimeter Institute for Theoretical Physics. Fabian Ballar Trigueros would like to thank the PSI program for facilitating this research. Research at Perimeter Institute is supported in part by the Government of Canada through the Department of

Innovation, Science and Economic Development Canada and by the Province of Ontario through the Ministry of Colleges and Universities.

# A  Gap ratio statistics of the random tilted Ising model

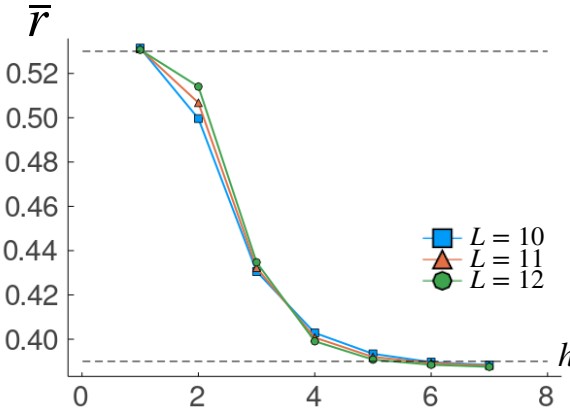

Figure 14: The gap ratio statistics $r$ for various disorder strengths $h$ and system sizes $L$. Note that the dashed lines corresponding to the mean values from a Gaussian orthogonal ensemble ($\overline{r} \approx 0.53$) and a Poisson ensemble ($\overline{r} \approx 0.39$). We therefore see that the system is in the MBL regime when $h \gtrsim 3$.

In this appendix, we show the gap ratio statistics [37] for the random-field quantum Ising model Eq. (12). In particular, we define $r = \min(\delta_n, \delta_{n-1})/\max(\delta_n, \delta_{n-1})$, where $\delta_n = E_{n+1} - E_n$ is the energy difference between two consecutive eigenstates. We take the middle half of the spectrum to calculate the mean value of $r$, and then average over $10^3$ disorder realizations to obtain $\overline{r}$ for various disorder strengths $h$ and system sizes $L$, plotted in Fig. (14). Note that for the Gaussian orthogonal ensemble, one expects $\overline{r} \approx 0.53$, while one expects $\overline{r} \approx 0.39$ for the Poisson distribution. We therefore see that $h \gtrsim 3$ is the regime where the system is in the many-body localization regime for the Hamiltonian Eq. (12).

# B  Additional data of Lanczos coefficients

In this appendix, we show additional data of the Lanczos coefficients. Particularly, we show several disorder realizations of $b_n$ from $L = 12$ and $h = 1.5$ (ergodic regime) initialized with $Z_0$ in Fig. 15, while show $b_n$ from $L = 12$ and $h = 7$ (MBL regime) in Fig. 16. In addition, motivated by the results of the zero mode in Sec. 3.4, we also show statistics of $\ln(b_{2n-1}/b_{2n})$ in Figs. 17 and 18 for the disorder strength $h = 1.5$ (ergodic regime) and $h = 7$ (MBL regime), respectively. Our results show that $\ln(b_{2n-1}/b_{2n})$ appear to be a normal distribution with a likely negative mean.

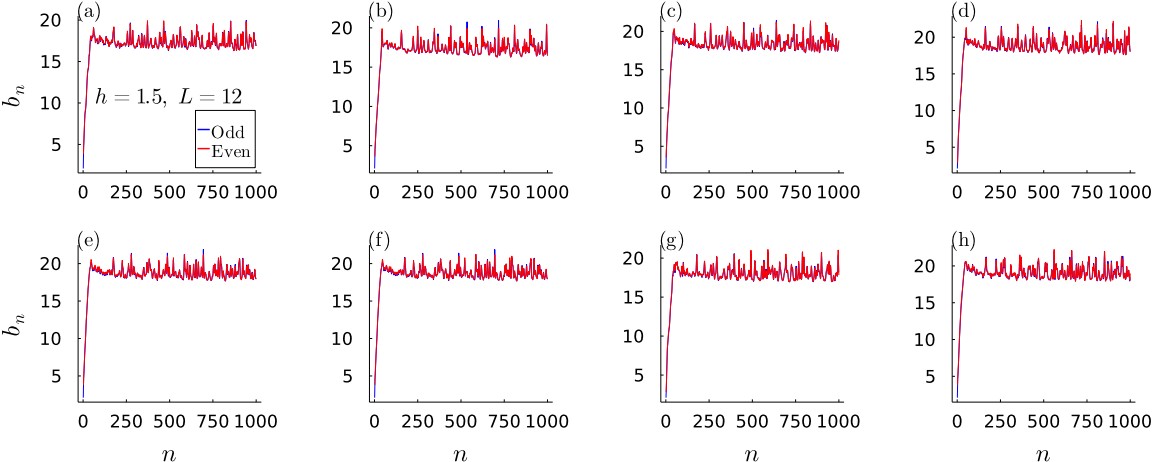

Figure 15: (a)-(h)Disorder realizations of the Lanczos coefficients $b_n$ for $L = 12$ and $h = 1.5$ (ergodic regime).

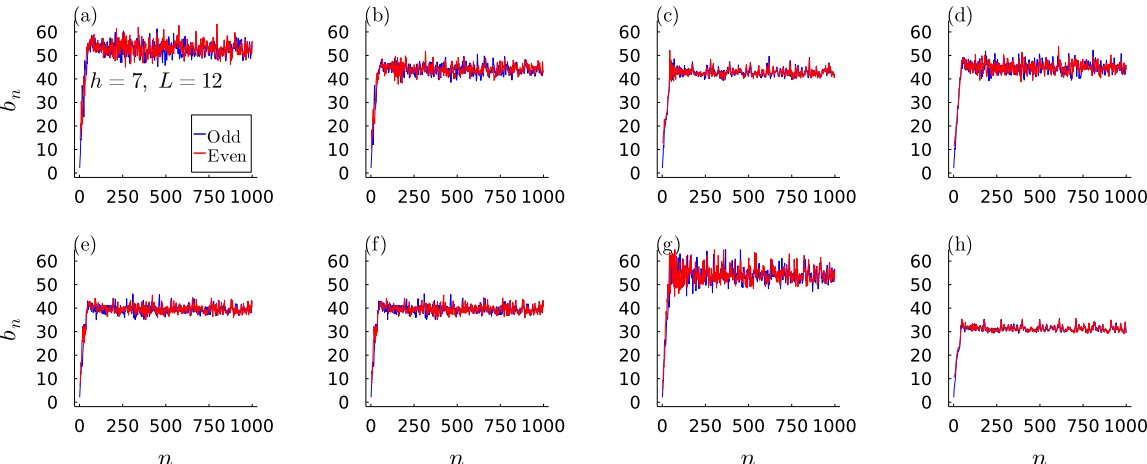

Figure 16: (a)-(h)Disorder realizations of the Lanczos coefficients $b_n$ for $L = 12$ and $h = 7$ (MBL regime).

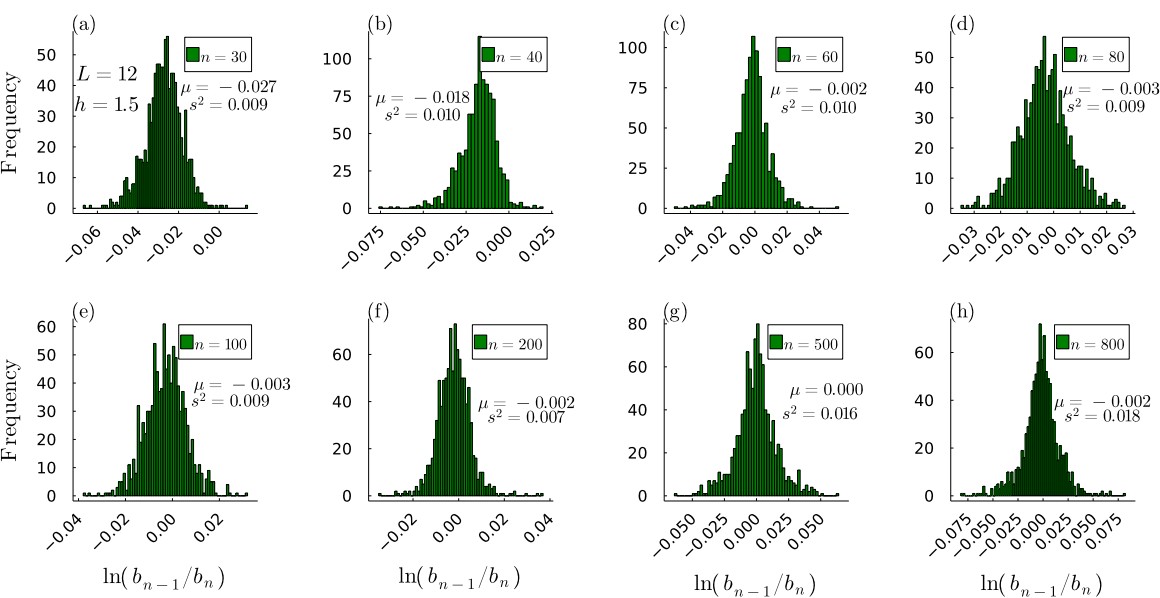

Figure 17: (a)-(h)Statistics of $\ln(b_{2n-1}/b_{2n})$ for several $n$, $L = 12$ and $h = 1.5$ (ergodic regime).

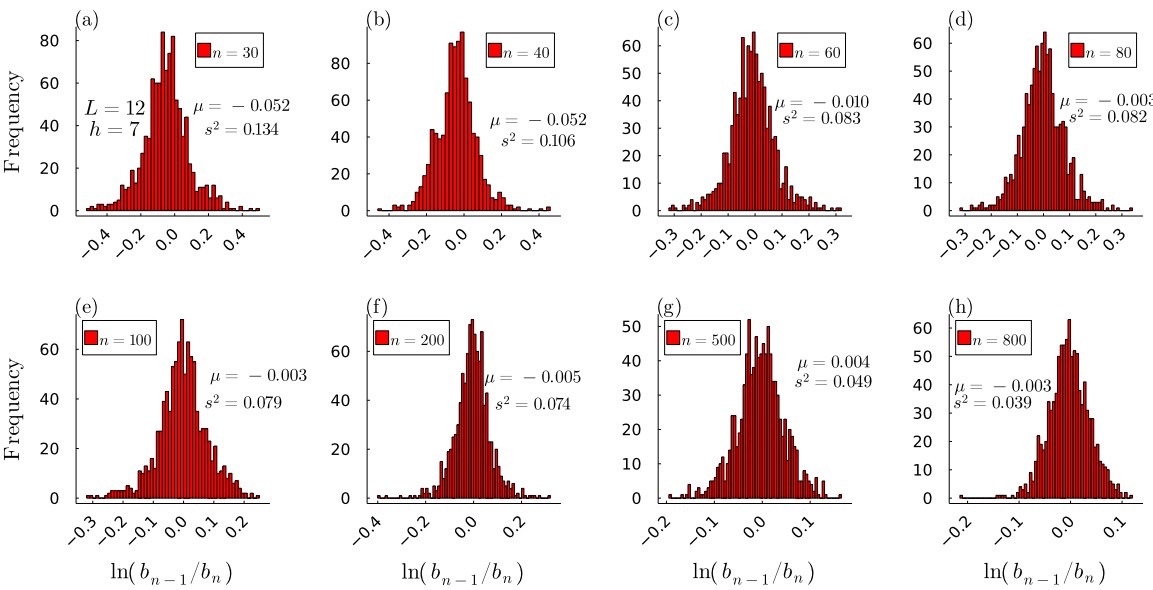

Figure 18: (a)-(h)Statistics of $\ln(b_{2n-1}/b_{2n})$ for several $n$, $L = 12$ and $h = 7$ (MBL regime).

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
