# Peer review of "Krylov complexity of many-body localization: Operator localization in Krylov basis"

_SciPost Physics_

## Round 1 · Referee Report · Anonymous (Referee 1) · 2022-2-20

Strengths

  1. The authors study an interesting and open question in the subfield.

Weaknesses

  1. Some of the central claims do no have sufficiently detailed numerical support.
  2. There are more typos than there should be.

Report

This work investigates the Operator Krylov Method (OKM) -- specifically a recent formulation of it in [25]-- in many-body localized systems. It's an interesting question, as OKM makes the assumption that operator lanczos coefficients grow linearly asymptotically. While this assumption appears to hold for ergodic systems, it's unclear how it is modified in the presence of disorder, and therefore it's unclear whether OKM is useful in these systems. [44] suggests that MBL appears as a subleading correction to the asymptotic form of the Lanczos coefficients.

The present paper investigates this issue further. They appear to find (numerically) that the lanczos coefficients (while growing linearly) have an even-odd modulation as well as an O(1) random fluctuation. This leads to an interesting conclusion: When the Liouvillian is expressed in the Lanczos basis, it ends up resembling a disordered single-particle hopping problem. The authors suggest that MBL corresponds to localization in this single particle problem, which is indeed an interesting suggestion.

While this manuscript likely meets the acceptance criteria (groundbreaking theoretical discovery), the authors will have to do more work to substantiate their main numerical claims.

I'll discuss the major physics issues here, and list typos/smaller issues in the requested changes.

First, the central claim (MBL corresponds to localization in Lanczos basis). The central numerical evidence for this claim is Fig 5-7. While these figures are quite compelling, it's unclear how well they really support the underlying hypothesis. For example, to be sure that the plateau in fig 6 correspond to localization, we need to see that the source of the plateau isn't simply the smallness of the system size. The authors should make clear the system sizes used for each figure, and also report how the results change with system size (perhaps in an appendix).

Second, the authors report that in the phenomenological model, lanczos coefficients saturate. As it stands this is a somewhat unilluminating part of the work. Do the lanczos coefficients saturate because the authors initial operator is an l-bit? Or does the lanczos coefficients saturate because the phenomenological model is limited (and only includes 2 body lbit operators)?

Requested changes

  1. Report system sizes in all figure captions.
  2. Provide data on convergence with system size in fig 6 at least, and add a discussion of this point to the text.
  3. Clarify reason for differences between phenomenological model and MBL model (see report).
  4. The last paragraph of the conclusion is very difficult to follow. The authors should clarify what they mean, and provide examples in the literature of "using computation complexity to characterise different phases of matter".
  5. Top of page 3: "to mention" -> "mentioning"
  6. Caption fig 5: "Noe" -> "Now"
  7. pg 9 "unextrpolated" -> "unextrapolated"

  • validity: good
  • significance: high
  • originality: high
  • clarity: good
  • formatting: reasonable
  • grammar: good

Author:  Cheng-Ju Lin  on 2022-06-20  [id 2595]

(in reply to Report 1 on 2022-02-20)

We would like to thank Referee for taking the time in reviewing our manuscript and providing us valuable comments and suggestions. Before answering and addressing the comments and questions raised by Referee, we would like to summarize the central idea of our major revision. In particular, in the context of the recent developments of MBL, there have been several works on challenging the existence of MBL in the thermodynamic limit. In this regard, we put more weights in this version on showing the operator localization in the MBL systems in the finite sizes, though we still keep the linear-extrapolation result in the manuscript as a simple-minded attempt to extrapolate it to the thermodynamic limit. Furthermore, in this version, we have included more numerical results with various system sizes to corroborate our claim of the operator localization in the finite sizes in the MBL systems.

Below, we address the comments and questions raised by Referee.

  1. We thank Referee for point out that our numerical results do not rule out that finite size is the reason of the plateauing of the Krylov complexity, and for requesting us to report the changes of Krylov complexity in various system sizes. In this version, we report the requested result in Fig. 8(a), which is calculated from the non-extrapolated Lanczos coefficients for several system sizes. As we see from the figure, in the MBL regime, the Krylov complexity indeed decreases when the system size increases from L = 10 to L = 12. We would also like to point out that such a behavior is in stark contrast with the Krylov complexity in the ergodic regime, which shows stretched exponential growth at the short time in Fig. 8(a). In order to further support our claim, we have also calculated the return probability (RP), reported in Fig.11 and in the last paragraph of Sec. 3.5. In particular, we attempt some simple finite-nmax and finite-L scaling. We observe that in the MBL regime, the RP retains a sizable value, which is indeed a hallmark of single-particle localization. We hope Referee would also agree that the results in Sec. 3.5 strongly suggest that the emergent single-particle problem is localized in the finite-size MBL system when initialized on the first site.

  2. Regarding Referee’s comment on the result of the MBL phenomenological model: Indeed that the different behavior of the Lanczos coefficients is simply due to the different choice of the initial operator hence the different Krylov/Lanczos basis. In our calculation, we include all the n-body interaction, so such a behavior is not coming from truncating the interactions in the model. We have mention and discuss this more explicitly in Sec.4 (see blue text). In addition, since this calculation is similar to the result of a single-spin dynamics, we add this comparison to the main text to give a better physical picture behind the calculation.

Requested changes: 1. We have modified the figures and the captions. 2. The additional data is reported in Fig. 8(a) and the discussion is in the second paragraph of Sec. 3.5. 3. Please see the blue-colored text in Sec. 4. 4. Please see the blue-colored text in Sec. 5, last paragraph. In particular, we are refereeing to the recent developments in understanding the intrinsic or symmetry protected topological phases of matter. One complexity characterization is through the circuit/state complexity: the obstruction of converting them into a product state with a (symmetry-preserved) finite-depth circuit. We also point out that there are recent works in associating the complexity of Monte Carlo calculation (namely, the existence of the sign problem) to the intrinsic property of some topological phases of matter. 5. We have fixed various spelling and grammar errors.

Attachment:

Summary_of_changes_t2CPOSk.pdf

---

## Round 1 · Referee Report · Anonymous (Referee 2) · 2022-3-19

Strengths

The topic of the paper: a study of systems with disorder and interactions in general, and many body localized (MBL) systems in particular, is very interesting and timely. Moreover the use of the Krylov method to the study of these systems, is a promising endeavor.

Weaknesses

  1. Poor citation of past literature which has discussed the even-odd effect and its influence on operator spreading.
  2. Unconvincing interpretation of the numerical results.

Report

The paper studies MBL systems using the method of mapping operator dynamics to single particle dynamics in a Krylov subspace. By studying the hopping parameters bn of the Krylov chain, the authors argue that while the overall behavior of the bn are very similar to chaotic systems by being n/ln(n), the bn also show an even odd effect which is responsible for localizing the particle in Krylov subspace. I have the following comments:

  1. The effect of the even-odd effect on localization is not new. The authors should make this fact clear when discussing past literature. The book by Vishwanath that the authors cite, discusses even-odd effects. The two papers by Yates et al that the authors cite, also discuss even odd effects in detail.

  2. The papers by Yates et al showed that when there is an even odd effect superimposed on a linear slope, the particle on the first site of the Krylov chain, stays localized. However if the even-odd effect persists only up to a finite distance into the chain, it makes the operator quasi-stable, causing the operator to decay into the bulk. These decay times can be very long. Looking at the authors plots, it appears they have a quasi-stable mode at the end of the Krylov chain rather than a stable mode as their even-odd effect does not persist into the bulk. Moreover the point where the even-odd effects terminate appears to be system size independent. So I am not convinced that their results imply a delta function peak in their low frequency spectral weight. The authors should discuss this physics and consider a more careful interpretation of their results.

This is also important as recent papers by Sels et al and Vidmar et al, have raised questions about past numerical metrics for MBL and there is new debate again on absolute stability vs quasi-stability.

In summary the paper presents solid numerical data, but I have concerns about the physical interpretation.

Requested changes

  1. Proper citation of past literature in the context of the even-odd effect in Krylov chains.
  2. A more careful discussion of their numerical results especially concerning the finite extent of the even-odd effects in their Krylov chain.
  3. A more up-to-date citation to numerical studies of MBL such as recent studies by Sels et al and Vidmar et al.

  • validity: ok
  • significance: ok
  • originality: low
  • clarity: ok
  • formatting: reasonable
  • grammar: reasonable

Author:  Cheng-Ju Lin  on 2022-06-20  [id 2594]

(in reply to Report 2 on 2022-03-19)

We would like to thank Referee for taking the time in reviewing our manuscript and providing us valuable comments and suggestions. Before answering and addressing the comments and questions raised by Referee, we would like to summarize the central idea of our major revision. In particular, in the context of the recent developments of MBL, there have been several works in challenging the existence of MBL in the thermodynamic limit. In this regard, we put more weights in this version on showing the operator localization in the MBL systems in the finite-sizes, though we still keep the linear-extrapolation result in the manuscript as a simple-minded attempt to extrapolate it to the thermodynamic limit. Furthermore, in this version, we have included more numerical results with various system sizes to corroborate our claim of the operator localization in the finite sizes in the MBL systems.

Below, we address the comments and questions raised by Referee.

  1. Indeed as Referee pointed out, there are various models whose Lanczos coefficients also show even-odd alteration. We have added more discussion about this effect in the last paragraph of Sec. 3.1 and in the last paragraph of Sec. 3.4. (Please see the blue-colored text.) However, we would like to point out that in our case, the even-odd alteration is hidden behind an apparent randomness, as shown in Fig. 2. This is a distinctive feature of our result, and could result in behaviors qualitatively different from the case of clean even-odd alteration.

  2. We believe what Referee is alluding to is the localization of the “zero mode”, which we did not address in detail in our previous version. Therefore, motivated by Referee’s comment, we study the zero mode in more details as shown in Sec. 3.4 and Fig. 7. However, our results seem to suggest localization of the zero mode on the first site, which could be a consequence of the apparent randomness. Furthermore, in Fig. 5, we attempt a finite-nmax and finite-L scaling of the amplitude of the zero-frequency delta function. We also compare the results with the results in ED. Our numerical results suggest that the zero-frequency delta function is indeed present at least in finite sizes, and also likely present in the thermodynamic limit. On the other hand, our main claim is the localization of the particle in the dynamics. That is, if a particle is initialized on the first site, then it will not propagate out and will stay localized around the first site. This is independent of the localization of the zero mode, though there could be some relations. And our results in Sec. 3.5 strongly suggest the aforementioned localization in the finite- size MBL systems. In particular, the results of the suppressed Krylov complexity, hardly-moving wavefunction profile in times and sizable return probability in Sec. 3.5 are strong indications. With these additional results and discussions in the main text, we hope Referee would also agree that our numerical results support our claim.

  3. We thank Referee for point out the references and the recent developments in the MBL research. Indeed, we have added the second last paragraph to the introduction and included various references including the ones suggested by Referee. As mentioned in the beginning, accordingly, we also put more emphasis on showing the localization of the operator in the finite-size MBL systems.

Requested changes: 1. We have added more discussions and citations for the past literature in Sec.3.1 last paragraph and Sec. 3.4 last paragraph. 2. Please see the additional results (blue-colored text) in Sec. 3. 3. Please see the blue-colored text in Introduction (second last paragraph).

Attachment:

Summary_of_changes.pdf

---

## Editorial Decision

resubmitted